# SD-LoRA: Scalable Decoupled Low-Rank Adaptation for Class Incremental Learning

**Yichen Wu**[1,2]*, **Hongming Piao**[1]*, **Long-Kai Huang**[4]†, **Renzhen Wang**[3], **Wanhua Li**[2],
**Hanspeter Pfister**[2], **Deyu Meng**[3,6], **Kede Ma**[1]†, **Ying Wei**[5]†
[1]City University of Hong Kong, [2]Harvard University, [3]Xi'an Jiaotong University,
[4]Tencent AI Lab, [5]Zhejiang University, [6]Pengcheng Laboratory

## Abstract

Continual Learning (CL) with foundation models has recently emerged as a promising paradigm to exploit abundant knowledge acquired during pre-training for tackling sequential tasks. However, existing prompt-based and Low-Rank Adaptation-based (LoRA-based) methods often require expanding a prompt/LoRA pool or retaining samples of previous tasks, which poses significant scalability challenges as the number of tasks grows. To address these limitations, we propose Scalable Decoupled LoRA (SD-LoRA) for class incremental learning, which continually separates the learning of the magnitude and direction of LoRA components without rehearsal. Our empirical and theoretical analysis reveals that SD-LoRA tends to follow a low-loss trajectory and converges to an overlapping low-loss region for all learned tasks, resulting in an excellent stability-plasticity trade-off. Building upon these insights, we introduce two variants of SD-LoRA with further improved parameter efficiency. All parameters of SD-LoRAs can be end-to-end optimized for CL objectives. Meanwhile, they support efficient inference by allowing direct evaluation with the finally trained model, obviating the need for component selection. Extensive experiments across multiple CL benchmarks and foundation models consistently validate the effectiveness of SD-LoRA. The code is available at `https://github.com/WuYichen-97/SD-Lora-CL`.

## 1 Introduction

Continual Learning (CL, Rolnick et al. 2019, Wang et al. 2024b, Zhou et al. 2024, Wang et al. 2022b) aims to develop computational learning systems capable of continually adapting to evolving environments while retaining previously acquired knowledge. In contrast to standard supervised learning, which assumes that training data are independent and identically distributed (i.i.d.), CL trains models on non-stationary data where tasks are presented sequentially. This departure from the i.i.d. assumption introduces the central challenge of *catastrophic forgetting* (French, 1999; McClelland et al., 1995; McCloskey & Cohen, 1989; Kirkpatrick et al., 2017), indicated by a significant performance degradation of previous tasks when new tasks are introduced.

Over the last five years, foundation models—large-scale pre-trained neural networks—have proven highly effective at transferring knowledge while exhibiting strong resistance to catastrophic forgetting (Wang et al., 2022b;a; Smith et al., 2023; Huang et al., 2024; Wang et al., 2024a; Liang & Li, 2024) in the context of CL. One prominent approach centers on adapting *input and intermediate* representations (*i.e*, prompts) of foundation models to accommodate new tasks. For example, L2P (Wang et al., 2022b) and DualPrompt (Wang et al., 2022a) incrementally learn a prompt pool and selectively insert the most relevant prompts based on their match with incoming test samples. CODA-Prompt (Smith et al., 2023) refines this strategy by end-to-end optimizing the prompt selection module for CL objectives. Despite obviating manual task identifiers, these methods rely heavily on accurately identifying task-relevant prompts from a potentially growing pool, raising concerns about inference scalability.

---

*Equal contribution.
†Corresponding authors.

Table 1: Comparisons of existing CL methods with foundation models in terms of three desirable properties: 1) *Rehearsal-free* (*i.e*, without memory for sample storage), 2) *inference efficiency* (*i.e*, without additional computational overhead during inference), and 3) *end-to-end optimization* (of all model parameters for CL objectives).

| Method | Rehearsal-free | Inference Efficiency | End-to-end Optimization |
|---|---|---|---|
| L2P (Wang et al., 2022b) | ✓ | ✗ | ✗ |
| DualPrompt (Wang et al., 2022a) | ✓ | ✗ | ✗ |
| CODA-Prompt (Smith et al., 2023) | ✓ | ✗ | ✓ |
| HiDe-Prompt (Wang et al., 2024a) | ✗ | ✗ | ✓ |
| InfLoRA (Liang & Li, 2024) | ✗ | ✓ | ✓ |
| SD-LoRA(Ours) | ✓ | ✓ | ✓ |

Another line of work has taken a more memory-intensive route by retaining samples of previous tasks to bolster performance. Building upon CODA-Prompt, HiDe-Prompt (Wang et al., 2024a) continues to learn prompts incrementally but stores large quantities of samples. Likewise, InfLoRA (Liang & Li, 2024) leverages Low-Rank Adaptation (LoRA) (Hu et al., 2022) to remain parameter-efficient, yet similarly rehearses extensive samples during incremental LoRA optimization. This reliance on large-scale memory makes them less scalable in real-world deployments, particularly in resource-constrained or large-scale CL settings.

Table 1 outlines three desirable properties for an ideal CL method with foundation models:

- **Rehearsal-free**: The method should eliminate the need to store samples from previous tasks, thereby ensuring learning scalability;
- **Inference Efficiency**: The method should maintain computational efficiency during inference, preferably without added computational costs, thereby ensuring inference scalability;
- **End-to-end Optimization**: All method parameters should be end-to-end optimized for CL objectives, rather than through segmented and separate optimization stages, thereby maximizing CL performance.

To achieve these properties, we propose Scalable and Decoupled LoRA (SD-LoRA), which incrementally adds LoRA components by separating the magnitude and direction learning. By directly employing the finally trained model for testing—without task-specific component selection—SD-LoRA supports computationally *efficient inference* (Huang et al., 2024), while being *rehearsal-free*. Through an in-depth empirical and theoretical analysis, we show that SD-LoRA learns to follow a low-loss path that converges to an overlapping low-loss region for all learned tasks, thus achieving an excellent stability-plasticity trade-off. Meanwhile, the importance of the incrementally learned LoRA directions diminishes as CL progresses. Building upon these observations, we introduce two variants of SD-LoRA with improved parameter efficiency through rank reduction and knowledge distillation, respectively. All SD-LoRA parameters can be *end-to-end optimized* for CL objectives.

In summary, the principal contributions of our work include

- A CL method with foundation models—SD-LoRA, offering a *rehearsal-free*, *inference-efficient*, and *end-to-end optimized* solution. We additionally include two SD-LoRA variants that improve parameter efficiency;
- An empirical and theoretical analysis of SD-LoRA, elucidating its plausible working mechanism that eliminates task-specific component selection;
- A comprehensive experimental evaluation of SD-LoRAs, demonstrating their effectiveness across multiple CL benchmarks and foundation models.

## 2 RELATED WORK

**Continual Learning (CL).** CL seeks to sequentially learn from new tasks while retaining previously acquired knowledge, with the central goal of mitigating catastrophic forgetting. Broadly, existing CL methods can be categorized according to three main design philosophies: Rehearsal-based, regularization-based, and architecture-based approaches. Rehearsal-based methods (Riemer et al.,

2018; Chaudhry et al., 2019; Tiwari et al., 2022) selectively retain and replay samples from previous tasks to alleviate catastrophic forgetting. Regularization-based methods (Kirkpatrick et al., 2017; Li & Hoiem, 2017; Lee et al., 2019) introduce penalty terms into the training objective to constrain updates on parameters deemed important for learned tasks. Architecture-based methods (Mallya et al., 2018; Ebrahimi et al., 2020; Ramesh & Chaudhari, 2021) expand or adapt the model architecture to account for new tasks. By allocating additional task-specific parameters or modules, these methods prevent the overwriting of learned important weights. Among various CL settings, this paper focuses on particularly challenging and practical class-incremental learning, in which the model must perform all learned tasks with no access to task identity at test time. Conventional class-incremental learning methods often require expensive training from scratch or parameter-intensive tuning, which can lead to overfitting and interference among tasks.

**CL with Foundation Models.** Foundation models have recently demonstrated their effectiveness in CL by facilitating knowledge transfer across tasks and reducing catastrophic forgetting (Wang et al., 2022b;a; Smith et al., 2023; Wang et al., 2024a; Liang & Li, 2024). Specifically, methods like L2P (Wang et al., 2022b), DualPrompt (Wang et al., 2022a), and CODA-Prompt (Smith et al., 2023) integrate Vision Transformers (ViTs) with prompt-tuning strategies (Lester et al., 2021; Jia et al., 2022), thereby improving knowledge retention as new tasks are introduced. Building on these, HiDe-Prompt (Wang et al., 2024a) further stores a large number of samples to boost performance. Rather than prompt-tuning, InfLoRA (Liang & Li, 2024) adopts a LoRA-based Parameter-Efficient Fine-Tuning (PEFT) approach, which similarly necessitates substantial sample storage. Despite their promise, none of the existing CL methods with foundation models simultaneously satisfy the three desirable properties outlined in Table 1. To fill this gap, we introduce SD-LoRA, which can also be viewed as a form of model-merging techniques (Chitale et al., 2023; Ilharco et al., 2023), developed in parallel with, yet complementary to ongoing CL research.

**Parameter-Efficient Fine-Tuning (PEFT).** The integration of PEFT methods with CL with foundation models is essential because full fine-tuning for each individual task is prohibitive in terms of computation and storage requirements. Representative PEFT methods include adapters (Houlsby et al., 2019), which insert lightweight learnable modules into Transformer layers; prompt-tuning (Qin & Eisner, 2021; Jia et al., 2022) and prefix-tuning (Li & Liang, 2021), which introduce learnable input representations into Transformer layers; and LoRA (Hu et al., 2022), which adds and tunes low-rank branches as updates to the pre-trained weights. While these techniques have proven effective in single-task and multi-task offline learning settings, their performance boundaries in CL with foundation models remain insufficiently explored. The proposed SD-LoRA adopts a rehearsal-free, LoRA-based PEFT approach for CL with foundation models.

## 3 PROPOSED METHOD: SD-LoRA

In this section, we first present the necessary preliminaries. We then present in detail the SD-LoRA method for CL with foundation models, accompanied by an empirical and theoretical analysis. Finally, we describe two SD-LoRA variants with improved parameter efficiency.

### 3.1 PRELIMINARIES

**Problem Formulation.** Let $\{\mathcal{T}_1, \mathcal{T}_2, \ldots, \mathcal{T}_N\}$ be $N$ sequential classification tasks. The training split of $\mathcal{T}_t$, denoted as $\mathcal{D}_t = \{\boldsymbol{x}_t^{(i)}, y_t^{(i)}\}_{i=1}^{|\mathcal{D}_t|}$, comprises of $|\mathcal{D}_t|$ training example pairs, where $\boldsymbol{x}_t^{(i)}$ representing the input image and $y_t^{(i)}$ its corresponding label. We consider a classification model $f_{\boldsymbol{\theta}}$, parameterized by $\boldsymbol{\theta}$. When training on $\mathcal{D}_t$, no data from previous tasks $\{\mathcal{T}_k\}_{k=1}^{t-1}$ is accessible. Accordingly, the training objective is given by

$$\ell\left(\mathcal{D}_t; \boldsymbol{\theta}\right) = \frac{1}{|\mathcal{D}_t|} \sum_{i=1}^{|\mathcal{D}_t|} \ell\left(f_{\boldsymbol{\theta}}\left(\boldsymbol{x}_t^{(i)}\right), y_t^{(i)}\right), \tag{1}$$

where $\ell(\cdot, \cdot)$ is a per-sample loss function such as cross-entropy. For model evaluation, we may compute the average loss of $f_{\boldsymbol{\theta}}$ across all tasks encountered so far: $\frac{1}{t} \sum_{k=1}^{t} \ell(\mathcal{V}_k; \boldsymbol{\theta})$, where $\mathcal{V}_k$ denotes the test split of $\mathcal{T}_k$. That is, the overarching goal is to ensure that $f_{\boldsymbol{\theta}}$ performs well on both the current task and all previous tasks. We generally follow the class-incremental learning setting described in (Wang et al., 2022b; Liang & Li, 2024).

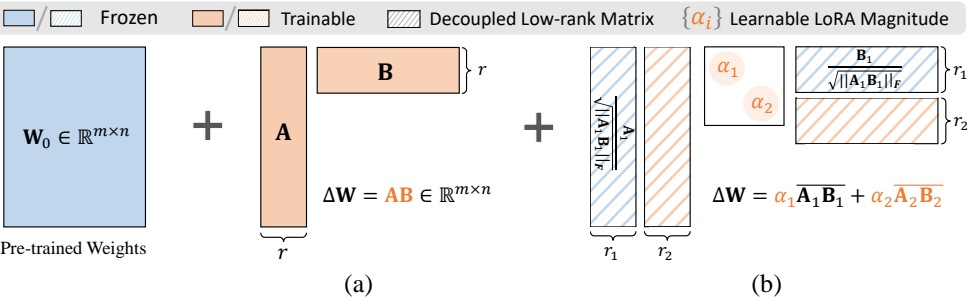

Figure 1: Illustration of the parameter update in **(a)** Vanilla LoRA and **(b)** the proposed SD-LoRA, where the current task index is $t = 2$ and $r, r_1, r_2 \ll \min\{m, n\}$.

**Low-Rank Adaptation (LoRA).** As illustrated in Fig. 1(a), LoRA (Hu et al., 2022) constrains the parameter updates during fine-tuning to lie in a low-rank subspace. Concretely, let $\mathbf{W}_0 \in \mathbb{R}^{m \times n}$ denote the original weight matrix of a layer in the classifier $f_{\boldsymbol{\theta}}$. LoRA expresses the parameter update $\Delta \mathbf{W} \in \mathbb{R}^{m \times n}$ as the product of two learnable matrices $\mathbf{A} \in \mathbb{R}^{m \times r}$ and $\mathbf{B} \in \mathbb{R}^{r \times n}$, *i.e.*, $\Delta \mathbf{W} = \mathbf{AB}$, with $r \ll \min\{m, n\}$. For a given layer of $f_{\boldsymbol{\theta}}$, the LoRA-updated output is

$$\boldsymbol{h}' = \mathbf{W}_0 \boldsymbol{x} + \Delta \mathbf{W} \boldsymbol{x} = (\mathbf{W}_0 + \mathbf{AB}) \boldsymbol{x}. \tag{2}$$

Throughout fine-tuning, the original weight matrix $\mathbf{W}_0$ remains fixed.

### 3.2 SD-LoRA

In LoRA, the parameter update $\Delta \mathbf{W}$ can be decomposed as follows:

$$\Delta \mathbf{W} = \|\mathbf{AB}\|_F \cdot \overline{\mathbf{AB}} = \|\mathbf{AB}\|_F \cdot \frac{\mathbf{AB}}{\|\mathbf{AB}\|_F}. \tag{3}$$

This decomposition highlights two crucial elements of the update: The *magnitude* (*i.e*, the Frobenius norm $\|\mathbf{AB}\|_F$) and *direction* (*i.e*, the normalized matrix $\overline{\mathbf{AB}}$). Recent work (Liu et al., 2024) has demonstrated that compared to full fine-tuning, LoRA exhibits limited flexibility in precisely adjusting these two elements. This drawback hinders its performance on complex tasks that demand fine-grained control over both magnitude and direction. Furthermore, Qiu et al. (2023) highlighted a more critical role of the direction in fine-tuning than the magnitude.

Motivated by these observations, we describe SD-LoRA for CL with foundation models. In a nutshell, SD-LoRA incrementally decouples the learning of the magnitude and direction of LoRA components, while fixing the directions learned from previous tasks as CL progresses. Concretely, let $\mathcal{M} = \{\alpha_k\}_{k=1}^t$ denote the learnable LoRA magnitudes, and $\mathcal{W} = \{\overline{\mathbf{A}_k \mathbf{B}_k}\}_{k=1}^{t-1}$ represent the previously learned directions. As illustrated in Fig. 1(b), during learning on $\mathcal{T}_t$, SD-LoRA computes the output of a given layer of the classifier $f_{\boldsymbol{\theta}}$ by

$$\boldsymbol{h}' = (\mathbf{W}_0 + \alpha_1 \overline{\mathbf{A}_1 \mathbf{B}_1} + \alpha_2 \overline{\mathbf{A}_2 \mathbf{B}_2} + \ldots + \alpha_t \overline{\mathbf{A}_t \mathbf{B}_t}) \boldsymbol{x}, \tag{4}$$

where the color-highlighted terms $\{\alpha_k\}_{k=1}^t$ and $\overline{\mathbf{A}_t \mathbf{B}_t}$ are learnable. The original weight matrix $\mathbf{W}_0$ and the previously learned directions $\{\overline{\mathbf{A}_k \mathbf{B}_k}\}_{k=1}^{t-1}$ remain fixed.

### 3.3 EMPIRICAL ANALYSIS OF SD-LoRA

By incrementally decoupling the magnitude and direction of LoRA components while preserving the directions learned from previous tasks, we observe substantial performance gains across various CL benchmarks (see Sec. 4). Nevertheless, the underlying working mechanism—particularly how SD-LoRA mitigates catastrophic forgetting—remains poorly understood. To shed light on this, we conduct a series of experiments and distill our observations into three key findings.

**Finding 1:** *When fine-tuning the foundation model directly on different downstream tasks, the resulting task-specific weights end up closer to each other than the original model weights.* To illustrate this, we consider five tasks drawn from ImageNet-R (Boschini et al., 2022), and fine-tune

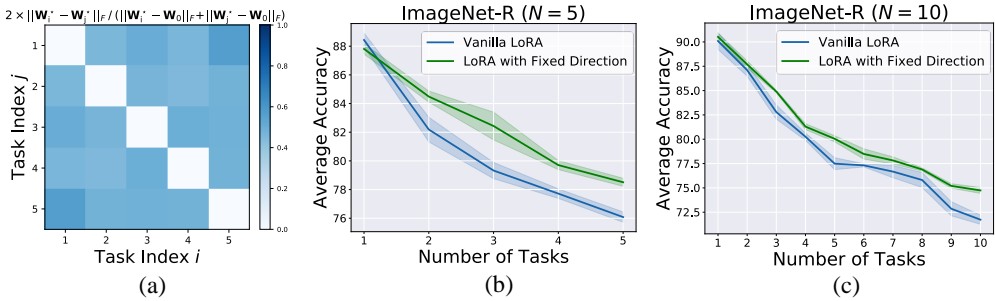

Figure 2: **(a)** Distances between the five optimal weights $\{\mathbf{W}_i^\star\}$ on ImageNet-R ($N = 5$) relative to the foundation model weights $\mathbf{W}_0$. All relative distances are much smaller than one, indicating that $\{\mathbf{W}_i^\star\}$ are closer to each other than to $\mathbf{W}_0$. **(b)** and **(c)** Performance comparison of Vanilla LoRA versus LoRA with the first learned direction fixed, on ImageNet-R across five and ten tasks, respectively. Shaded regions indicate standard error.

the ViT-B-16 model (Dosovitskiy et al., 2020) for each task, thereby obtaining five sets of optimal task-specific weights $\{\mathbf{W}_i^\star\}_{i=1}^5$. As shown in Fig. 2(a), measuring the relative distances in parameter space reveals that these task-specific weights cluster more closely with one another than the original weights of the foundation model $\mathbf{W}_0$.

We further conduct a CL experiment in which only the LoRA magnitude is continually optimized, while the direction remains fixed after the first task. Consequently, the updated output for a given layer at the current $\mathcal{T}_t$ becomes $h' = (\mathbf{W}_0 + \alpha \overline{\mathbf{A}_1 \mathbf{B}_1})x$. As shown in Figs. 2(b) and (c), the average accuracy up to the current task consistently surpasses that of the vanilla LoRA baseline, *i.e*, $h' = (\mathbf{W}_0 + \mathbf{AB})x$. Aligning well with (Entezari et al., 2022; Gueta et al., 2023), our results further indicate that the fine-tuned weights for different tasks lie in close proximity, enabling relatively strong performance even when fixing a single learned direction.

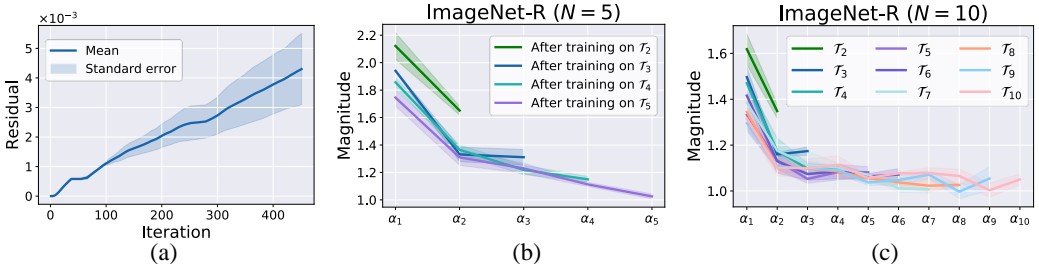

Figure 3: Analysis of the learning process of SD-LoRA. **(a)** Least squares fitting residual between the newly learned direction $\overline{\mathbf{A}_t \mathbf{B}_t}$ and all previous directions $\{\overline{\mathbf{A}_k \mathbf{B}_k}\}_{k=1}^{t-1}$ over time. **(b)** and **(c)** Learned magnitudes $\{\alpha_k\}_{k=1}^N$ on ImageNet-R across five and ten tasks, respectively.

**Finding 2:** *The directions preserved from previous tasks (i.e., $\{\overline{\mathbf{A}_k \mathbf{B}_k}\}_{k=1}^{t-1}$) play a significant role in the CL process—particularly those learned in the initial tasks.* To reveal this, we first compute the least squares fitting residual between $\overline{\mathbf{A}_t \mathbf{B}_t}$ and $\{\overline{\mathbf{A}_k \mathbf{B}_k}\}_{k=1}^{t-1}$, which increases over time (see Fig. 3(a)). Initially, the newly learned direction strongly aligns with earlier ones, enabling learned direction reuse. As training continues, $\overline{\mathbf{A}_t \mathbf{B}_t}$ gradually diverges, incorporating subtle variations that distinguish it from previous directions.

Further analysis of the learned magnitudes, all initialized to ones, reveals that $\alpha_k$ values corresponding to earlier tasks rise rapidly, while those for later tasks exhibit a general decline trend (see Figs. 3(b) and (c) as well as Appendix A.2). This pattern suggests that the classifier increasingly relies on directions learned from the earlier tasks, whereas the recently introduced directions serve primarily as slight adjustments to accommodate the specific requirements of the later tasks.

**Finding 3:** *SD-LoRA effectively uncovers a low-loss path by leveraging the fixed directions from previous tasks with the learned magnitudes $\{\alpha_k\}_{k=1}^N$, toward a low-loss region shared by all tasks.* To

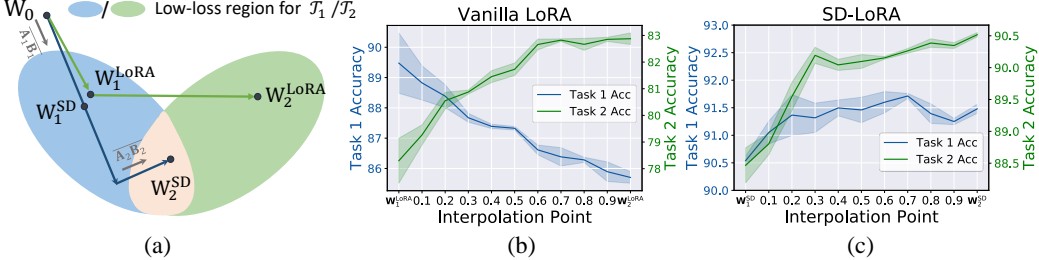

(a)          (b)          (c)

Figure 4: Learning trajectory comparison of vanilla LoRA and SD-LoRA. **(a)** Toy illustration of the learning trajectories for vanilla LoRA ($\mathbf{W}_0 \rightarrow \mathbf{W}_1^{\text{LoRA}} \rightarrow \mathbf{W}_2^{\text{LoRA}}$) and SD-LoRA ($\mathbf{W}_0 \rightarrow \mathbf{W}_1^{\text{SD}} \rightarrow \mathbf{W}_2^{\text{SD}}$) across two sequential tasks. **(b)** Classification accuracy along the vanilla LoRA path. The improvement on $\mathcal{T}_2$ but degradation on $\mathcal{T}_1$ indicates that vanilla LoRA suffers from catastrophic forgetting. **(c)** Classification accuracy along the SD-LoRA path, showing that it successfully lands on an overlapping low-loss region.

verify this, we conduct weight interpolation experiments to examine the linear path between the two sets of model weights for two sequential tasks. As depicted in Fig. 4, along the linear path from $\mathbf{W}_1^{\text{SD}}$ to $\mathbf{W}_2^{\text{SD}}$ learned by SD-LoRA, the performance on $\mathcal{T}_2$ steadily improves without loss in accuracy on $\mathcal{T}_1$. However, this is not the case for vanilla LoRA, where the performance improvement of $\mathcal{T}_2$ is at the expense of $\mathcal{T}_1$, indicative of catastrophic forgetting. These observations suggest that SD-LoRA selectively scales the parameter update along the previously learned directions, effectively enabling the classifier to trace a low-loss path that ultimately settles on an overlapping low-loss region for all tasks (see Fig. 4 (a)).

These findings elucidate why SD-LoRA excels in CL with foundation models. Initially, it identifies critical LoRA directions during learning earlier tasks, and relies heavily on these directions to guide the classifier toward an overlapping low-loss region for learned tasks. Subsequently, by progressively incorporating LoRA components, SD-LoRA refines these directions to converge on the shared low-loss region for both earlier and later tasks. This mechanism of tracing a low-loss trajectory eliminates the need to store samples from previous tasks for task-specific component selection, making SD-LoRA strong and rehearsal-free.

### 3.4 THEORETICAL ANALYSIS OF SD-LoRA

In this subsection, based on the results in (Jiang et al., 2023), we present a theoretical analysis to explain why the initially learned LoRA directions are so critical (as in Finding 2).

Let $\Delta \mathbf{W}^\star \in \mathbb{R}^{m \times n}$ be the optimal update matrix lying in the overlapping low-loss region for all $N$ sequential tasks. Additionally, let $\{\Delta \mathbf{W}_i^\star\}_{i=1}^N$ represent the optimal update matrices in their respective low-loss regions. Denote the singular values of $\Delta \mathbf{W}_t^\star$ as $\sigma_1 \geq \ldots \geq \sigma_{\min\{m,n\}} \geq 0$. The matrices $\mathbf{A} \in \mathbb{R}^{m \times r}$ and $\mathbf{B} \in \mathbb{R}^{r \times n}$ are updated iteratively, starting from initial values given by $\frac{\rho}{3\sqrt{m+n+r}}(\mathbf{A}_0, \mathbf{B}_0)$, where the entries of $\mathbf{A}_0$ and $\mathbf{B}_0$ are i.i.d. according to $\mathcal{N}(0, \sigma_1)$, and $\rho$ is the initialization scaling factor. For an integer $j$ in the range $\{0, 1, \ldots, \min\{r, m, n\}\}$, define the $j$-th condition number as $\kappa_j = \frac{\sigma_1}{\sigma_j}$. Finally, let $\|\cdot\|_{\text{op}}$ denote the operator norm.

**Theorem 1.** *Suppose the assumptions stated in Appendix A.1 hold, where $\epsilon_1$ is a small constant. Let $\delta \in (0, 1)$ be such that $\delta \leq \min_{k \in \{1, \ldots, j\}} \frac{\sigma_k - \sigma_{k+1}}{\sigma_k}$. Fix any tolerance level $\epsilon_2$ satisfying $\epsilon_2 \leq \frac{1}{m+n+r}$. Let $\eta$ denote the learning rate for updating the matrices $\mathbf{A}$ and $\mathbf{B}$, and define $\Delta \mathbf{W}^{[:i]}$ as the rank-$i$ approximation of $\Delta \mathbf{W}^\star$, obtained by retaining the top-$i$ principal components.*

*Then, there exist some numerical constants $c$ and $c'$, and a sequence of iteration indices:*

$$i_1 \leq i_2 \leq \ldots \leq i_j \leq \frac{c'}{\delta \eta \sigma_j} \log\left(\frac{\kappa_j}{\delta \epsilon_2}\right)$$

*such that, with high probability, gradient descent with step size* $\eta \le c \min\{\delta, 1 - \delta\} \frac{\sigma_j^2}{\sigma_1^3}$ *and initialization scaling factor* $\rho \le \left(\frac{c\delta\epsilon_2}{\kappa_j}\right)^{\frac{1}{c\delta}}$ *ensures that the approximation error satisfies*

$$\left\| \mathbf{A}_{i_k} \mathbf{B}_{i_k} - \Delta \mathbf{W}^{[:k]} \right\|_{\text{op}} \le \epsilon_2 \sigma_1 + \epsilon_1, \quad \forall k = 1, 2, \dots, j. \tag{5}$$

In Theorem 1, we formulate the learning process of SD-LoRA as a matrix factorization problem, and prove that gradient descent with small initialization drives the learned product $\mathbf{AB}$ to approximate the principal components of $\Delta \mathbf{W}^\star$, *i.e*, $\Delta \mathbf{W}^{[1]}, \Delta \mathbf{W}^{[:2]}, \dots, \Delta \mathbf{W}^{[:j]}$, sequentially. This theoretical insight explains the observed decreasing trend in the learned magnitudes, and further supports the feasibility of the subsequent parameter-efficient variants of SD-LoRA.

### 3.5 TWO VARIANTS OF SD-LORA

Our analysis in Sec. 3.3 reveals an interesting behavior of SD-LoRA: LoRA directions acquired during learning on earlier tasks are heavily reused and contribute substantially to classification accuracy. In contrast, directions learned through later tasks primarily serve as minor refinements, exhibiting a diminishing utility. We leverage this behavior and propose two variants of SD-LoRA with improved parameter efficiency, which integrate rank reduction and knowledge distillation, termed SD-LoRA-RR and SD-LoRA-KD, respectively.

**SD-LoRA-RR.** To mitigate incremental parameter expansion, we implement an empirical rank-reduction strategy for the learnable matrices $\mathbf{A}_t \in \mathbb{R}^{m \times r_t}$ and $\mathbf{B}_t \in \mathbb{R}^{r_t \times n}$ associated with later tasks. Specifically, the rank $r_t$ is reduced in a stepwise manner:

$$r_1 = r_2 = \dots > r_\mu = r_{\mu+1} = \dots > r_\nu = r_{\nu+1} = \dots = r_N, \tag{6}$$

where $\mu$ and $\nu$ are predefined task indices. The set of hyperparameters include $\{\mu, \nu, r_1, r_\mu, r_\nu\}$. This stepwise reduction ensures that later tasks, which contribute less, are encoded with lower-rank approximations, thereby curbing computational and memory overhead.

**SD-LoRA-KD.** While the rank-reduction strategy limits parameter expansion, each new task still grows parameters incrementally. To fully address this issue, we propose a knowledge distillation approach based on least squares. Our method evaluates whether a newly introduced LoRA direction $\overline{\mathbf{A}_t \mathbf{B}_t}$ can be linearly represented by the subspace spanned by previously learned directions $\{\overline{\mathbf{A}_k \mathbf{B}_k}\}_{k=1}^{t-1}$. If a sufficient linear approximation exists, the fitting coefficients will be absorbed into existing learned magnitudes $\mathcal{M}$ rather than expanding the direction set $\mathcal{W}$. Formally, after training on $\mathcal{T}_t$, we solve the least squares optimization problem:

$$\{\Delta\alpha_k\}_{k=1}^{t-1} = \underset{\{\alpha_k'\}_{k=1}^{t-1}}{\arg\min} \left\| \overline{\mathbf{A}_t \mathbf{B}_t} - \sum_{k=1}^{t-1} \alpha_k' \overline{\mathbf{A}_k \mathbf{B}_k} \right\|_F^2, \tag{7}$$

where $\Delta\alpha_k$ denotes the optimal coefficient for the $k$-th learned direction. If the fitting residual is less than a predefined threshold $\tau$, we assimilate the new direction by updating the first $t-1$ learned magnitudes from $\mathcal{T}_t$ as

$$\boldsymbol{h}' = \left( \mathbf{W}_0 + (\alpha_1 + \Delta\alpha_1)\overline{\mathbf{A}_1 \mathbf{B}_1} + (\alpha_2 + \Delta\alpha_2)\overline{\mathbf{A}_2 \mathbf{B}_2} + \dots + (\alpha_{t-1} + \Delta\alpha_{t-1})\overline{\mathbf{A}_{t-1} \mathbf{B}_{t-1}} \right) \boldsymbol{x}, \tag{8}$$

which prevents parameter expansion while preserving knowledge through coefficient fusion. The complete implementations of SD-LoRAs are detailed in Algorithm 1.

## 4 EXPERIMENTS

In this section, we first present the experimental setups, and then compare SD-LoRAs with state-of-the-art CL methods across multiple benchmarks and foundation models.

### 4.1 EXPERIMENTAL SETUPS

**Evaluation Benchmarks and Protocols.** Following (Gao et al., 2023; Liang & Li, 2024), we evaluate SD-LoRAs on three standard CL benchmarks: ImageNet-R (Boschini et al., 2022), ImageNet-A (Hendrycks et al., 2021), and DomainNet (Peng et al., 2019). Specifically, ImageNet-R consists

---

**Algorithm 1** SD-LoRA and its Variants on the Current Task $\mathcal{T}_t$

---

**Input:** Weight matrix from the foundation model $\mathbf{W}_0$, current task $\mathcal{T}_t$, learned LoRA directions from previous tasks $\mathcal{W} = \{\overline{\mathbf{A}_k \mathbf{B}_k}\}_{k=1}^{t-1}$, rank parameters $\{\mu, \nu, r_1, r_\mu, r_\nu\}$ in Eqn. (6), residual threshold $\tau$ for Eqn. (7), and maximum number of iterations MaxIter.

**Output:** Sets of learned LoRA magnitudes $\mathcal{M}$ and directions $\mathcal{W}$.

 1: Initialize $\mathbf{A}_t \in \mathbb{R}^{m \times r_t}$, $\mathbf{B}_t \in \mathbb{R}^{r_t \times n}$, and $\mathcal{M} = \{\alpha_k\}_{k=1}^t$
 2: **if** $t = \mu$ or $\nu$ **then**           */ Only for SD-LoRA-RR*
 3:      Reduce the lower dimension of $\mathbf{A}_t$ and $\mathbf{B}_t$ to $r_\mu$ or $r_\nu$
 4: **end if**
 5: **for** Iter $= 0$ to MaxIter **do**
 6:      Compute the cross-entropy loss on the current task $\mathcal{T}_t$ using Eqn. (4)
 7:      Update $\{\alpha_k\}_{k=1}^t$ and $\overline{\mathbf{A}_t \mathbf{B}_t}$ by minimizing Eqn. (1) using some stochastic optimizer
 8: **end for**
 9: $\mathcal{W} \leftarrow \mathcal{W} \bigcup \{\overline{\mathbf{A}_t \mathbf{B}_t}\}$
10: Solve Problem (7) to obtain the optimal fitting coefficients $\{\Delta \alpha_k\}_{k=1}^{t-1}$ */ Only for SD-LoRA-KD*
11: **if** the fitting residual $\left\| \overline{\mathbf{A}_t \mathbf{B}_t} - \sum_{k=1}^{t-1} \Delta \alpha_k \overline{\mathbf{A}_k \mathbf{B}_k} \right\|_F \leq \tau$ **then**
12:      $\mathcal{M} \leftarrow \{\alpha_k + \Delta \alpha_k\}_{k=1}^{t-1}$ and $\mathcal{W} \leftarrow \{\overline{\mathbf{A}_k \mathbf{B}_k}\}_{k=1}^{t-1}$

---

Table 2: Performance comparison on ImageNet-R across different task lengths.

| Method | ImageNet-R ($N=5$) | | ImageNet-R ($N=10$) | | ImageNet-R ($N=20$) | |
|---|---|---|---|---|---|---|
| | Acc ↑ | AAA↑ | Acc ↑ | AAA ↑ | Acc ↑ | AAA ↑ |
| Full Fine-Tuning | $64.92_{(0.87)}$ | $75.57_{(0.50)}$ | $60.57_{(1.06)}$ | $72.31_{(1.09)}$ | $49.95_{(1.31)}$ | $65.32_{(0.84)}$ |
| L2P | $73.04_{(0.71)}$ | $76.94_{(0.41)}$ | $71.26_{(0.44)}$ | $76.13_{(0.46)}$ | $68.97_{(0.51)}$ | $74.16_{(0.32)}$ |
| DualPrompt | $69.99_{(0.57)}$ | $72.24_{(0.41)}$ | $68.22_{(0.20)}$ | $73.81_{(0.39)}$ | $65.23_{(0.45)}$ | $71.30_{(0.16)}$ |
| CODA-Prompt | $76.63_{(0.27)}$ | $80.30_{(0.28)}$ | $74.05_{(0.41)}$ | $78.14_{(0.39)}$ | $69.38_{(0.33)}$ | $73.95_{(0.63)}$ |
| HiDe-Prompt | $74.77_{(0.25)}$ | $78.15_{(0.24)}$ | $74.65_{(0.14)}$ | $78.46_{(0.18)}$ | $73.59_{(0.19)}$ | $77.93_{(0.19)}$ |
| InfLoRA | $76.95_{(0.23)}$ | $81.81_{(0.14)}$ | $74.75_{(0.64)}$ | $80.67_{(0.55)}$ | $69.89_{(0.56)}$ | $76.68_{(0.57)}$ |
| SD-LoRA | $\mathbf{79.15}_{(0.20)}$ | $\mathbf{83.01}_{(0.42)}$ | $\mathbf{77.34}_{(0.35)}$ | $\mathbf{82.04}_{(0.24)}$ | $\mathbf{75.26}_{(0.37)}$ | $80.22_{(0.72)}$ |
| SD-LoRA-RR | $79.01_{(0.26)}$ | $82.50_{(0.38)}$ | $77.18_{(0.39)}$ | $81.74_{(0.24)}$ | $74.05_{(0.51)}$ | $\mathbf{80.65}_{(0.35)}$ |
| SD-LoRA-KD | $78.85_{(0.29)}$ | $82.47_{(0.58)}$ | $77.03_{(0.67)}$ | $81.52_{(0.26)}$ | $74.12_{(0.66)}$ | $80.11_{(0.75)}$ |

of 200 ImageNet classes (Deng et al., 2009) rendered in artistic styles. ImageNet-A features 200 classes with natural adversarial examples, often misclassified by standard ImageNet-trained models. DomainNet includes 345 classes across six distinct domains. As common practices (Liang & Li, 2024; Huang et al., 2024), we split ImageNet-R into 5/10/20 tasks (40/20/10 classes per task), ImageNet-A into 10 tasks (20 classes each), and DomainNet into 5 tasks (69 classes each). Additionally, we include CIFAR100 (Krizhevsky, 2009) and CUB200 (Wah et al., 2011) results in Appendix A.3.

We adopt two standard and widely used CL metrics: Average accuracy (Acc) and average anytime accuracy (AAA). The Acc metric measures the overall performance by computing the average accuracy across all $N$ tasks upon the completion of CL. AAA further accumulates the average accuracy of all encountered tasks after training on each new task.

**Competing Methods and Implementation Details.** We compare SD-LoRAs against state-of-the-art ViT-based CL methods, including L2P (Wang et al., 2022b), DualPrompt (Wang et al., 2022a), CODA-Prompt (Smith et al., 2023), HiDe-Prompt (Wang et al., 2024a), and InfLoRA (Liang & Li, 2024). We also incorporate full fine-tuning as a form of performance lower bound. Following prior work (Gao et al., 2023; Huang et al., 2024), we employ ViT-B/16 (Dosovitskiy et al., 2020), pre-trained on ImageNet-21K and fine-tuned on ImageNet-1K as the foundation model for classification. We also experiment with a self-supervised ViT-B/16 from DINO (Caron et al., 2021). The SD-LoRA components are inserted into the attention layers of all Transformer blocks, modifying the query and value projections, with a fixed rank of $r_1 = 10$. It is noteworthy that we utilize a shared set of LoRA magnitudes for all projections, each with different LoRA directions. For SD-LoRA-RR, we set the additional rank parameters as $\mu = 4$, $\nu = 8$, $r_\mu = 8$, and $r_\nu = 6$. For SD-LoRA-KD, we set the threshold for the fitting residual to $\tau = 9 \times 10^{-4}$. For all methods, training is carried out by

Table 3: Performance comparison on ImageNet-A and DomainNet across different task lengths.

| Method | ImageNet-A ($N = 10$) | | DomainNet ($N = 5$) | |
|---|---|---|---|---|
| | Acc ↑ | AAA ↑ | Acc ↑ | AAA ↑ |
| Full Fine-Tuning | $16.31_{(7.89)}$ | $30.04_{(13.18)}$ | $51.46_{(0.47)}$ | $67.08_{(1.13)}$ |
| L2P (Wang et al., 2022b) | $42.94_{(1.27)}$ | $51.40_{(1.95)}$ | $70.26_{(0.25)}$ | $75.83_{(0.98)}$ |
| DualPrompt (Wang et al., 2022a) | $45.49_{(0.96)}$ | $54.68_{(1.24)}$ | $68.26_{(0.90)}$ | $73.84_{(0.45)}$ |
| CODA-Prompt (Smith et al., 2023) | $45.36_{(0.78)}$ | $57.03_{(0.94)}$ | $70.58_{(0.53)}$ | $76.68_{(0.44)}$ |
| HiDe-Prompt (Wang et al., 2024a) | $42.70_{(0.60)}$ | $56.32_{(0.40)}$ | $72.20_{(0.08)}$ | $77.01_{(0.04)}$ |
| InfLoRA (Liang & Li, 2024) | $49.20_{(1.12)}$ | $60.92_{(0.61)}$ | $71.59_{(0.23)}$ | $78.29_{(0.50)}$ |
| SDLoRA | $\mathbf{55.96}_{(0.73)}$ | $\mathbf{64.95}_{(1.63)}$ | $\mathbf{72.82}_{(0.37)}$ | $\mathbf{78.89}_{(0.50)}$ |
| SD-LoRA-RR | $55.59_{(1.08)}$ | $64.59_{(1.91)}$ | $72.58_{(0.40)}$ | $78.79_{(0.78)}$ |
| SD-LoRA-KD | $54.24_{(1.12)}$ | $63.89_{(0.58)}$ | $72.15_{(0.50)}$ | $78.44_{(0.66)}$ |

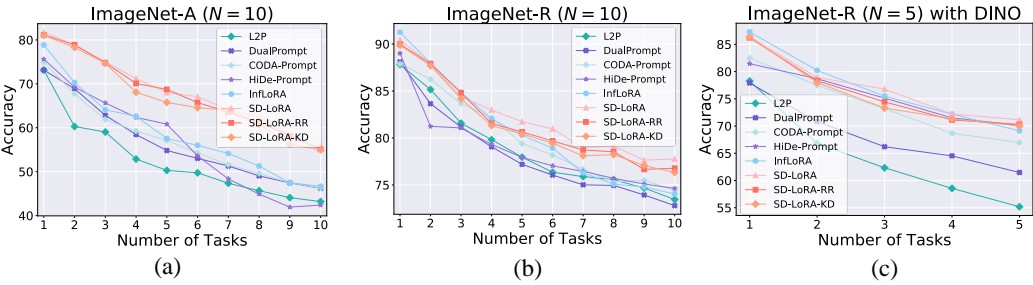

Figure 5: Average accuracy during sequential training on **(a)** ImageNet-A ($N = 10$), **(b)** ImageNet-R ($N = 10$), and **(c)** ImageNet-R ($N = 5$) using ViT-B/16 from DINO (Caron et al., 2021).

Adam (Kingma & Ba, 2014) with a learning rate of 0.008 and a minibatch size of 128 for 30 epochs on ImageNet-R, 10 epochs on DomainNet, and 20 epochs on all other datasets. We report mean results across five runs with standard errors.

## 4.2 EXPERIMENTAL RESULTS

**Results on Different CL benchmarks Using Different Backbones.** In Tables 2 and 3, it is clear that SD-LoRA achieves significant improvements over existing methods. Specifically, on ImageNet-R ($N = 20$), SD-LoRA surpasses InfLoRA by margins of $7.68\%$ in Acc and $4.62\%$ in AAA. Similarly, on ImageNet-A, SD-LoRA outperforms HiDe-prompt by approximately $31.05\%$ in Acc and $15.32\%$ in AAA. Even on the more complex DomainNet, comprising six distinct domains, SD-LoRA consistently attains the best performance. To demonstrate the generality of SD-LoRA, we also evaluate it using the self-supervised ViT-B/16 from DINO. The results in Fig. 5(c) show that SD-LoRA continues to deliver superior performance under both Acc and AAA using different backbones. Finally, we observe that the two variants SD-LoRA-RR and SD-LoRA-KD, exhibit only marginal performance degradations relative to the full SD-LoRA model, confirming the effectiveness of their parameter-efficient designs.

**Results across Varied Task Lengths.** To evaluate the scalability and generalizability of SD-LoRAs under different task lengths, we follow (Liang & Li, 2024; Huang et al., 2024) and partition ImageNet-R into 5, 10, and 20 sequential tasks containing 40, 20, and 10 classes per task, respectively. As shown in Table 2, SD-LoRAs demonstrate consistent superiority over existing methods, with performance margins growing as the number of tasks. This underscores the suitability of SD-LoRAs for scenarios requiring resource-efficient CL without compromising accuracy.

**Ablation Studies.** To validate the contributions of design choices in the proposed SD-LoRA, we conduct a series of ablation experiments, with quantitative results summarized in Table 4. First, we fix the singly learned LoRA direction while allowing its magnitude to adapt during training. This simplified configuration already achieves nontrivial performance, highlighting the critical role of the initial LoRA direction in CL. Second, we decouple the magnitude and direction learning, but restrict the classifier to a single LoRA component. The inferior performance relative to SD-LoRA

Table 4: Ablation analysis of the proposed SD-LoRA. Trainable parameters are highlighted in orange.

| Training Strategy | ImageNet-R ($N = 5$) | | ImageNet-R ($N = 10$) | |
|---|---|---|---|---|
| | Acc ↑ | AAA ↑ | Acc ↑ | AAA ↑ |
| $\mathbf{W}_0 + \alpha\overline{\mathbf{A}_1\mathbf{B}_1}$ | $78.17_{(0.27)}$ | $81.93_{(0.51)}$ | $74.82_{(0.96)}$ | $80.63_{(0.63)}$ |
| $\mathbf{W}_0 + \alpha\overline{\mathbf{AB}}$ | $73.24_{(0.31)}$ | $78.80_{(0.13)}$ | $70.62_{(0.78)}$ | $76.32_{(0.16)}$ |
| $\mathbf{W}_0 + \overline{\mathbf{A}_1\mathbf{B}_1} + \ldots + \alpha\overline{\mathbf{A}_t\mathbf{B}_t}$ | $78.28_{(0.59)}$ | $82.02_{(0.71)}$ | $74.29_{(0.32)}$ | $79.74_{(0.71)}$ |
| $\mathbf{W}_0 + \alpha_1\overline{\mathbf{A}_1\mathbf{B}_1} + \ldots + \alpha_t\overline{\mathbf{A}_t\mathbf{B}_t}$ (SD-LoRA) | $\mathbf{79.15}_{(0.20)}$ | $\mathbf{83.01}_{(0.42)}$ | $\mathbf{77.34}_{(0.35)}$ | $\mathbf{82.04}_{(0.24)}$ |

Table 5: Comparison on ImageNet-R ($N = 20$) in terms of computation (GFLOPs), parameter, and storage efficiency.

| Method | GFLOPs | Learnable Parameters (M) | Stored Features (M) |
|---|---|---|---|
| L2P (Wang et al., 2022b) | 70.14 | 0.48 | 0 |
| DualPrompt (Wang et al., 2022a) | 70.26 | 0.06 | 0 |
| CODA-Prompt (Smith et al., 2023) | 70.61 | 0.38 | 0 |
| HiDe-Prompt (Wang et al., 2024a) | 70.36 | 0.08 | 0.15 |
| InfLoRA (Liang & Li, 2024) | 35.12 | 0.37 | 0.10 |
| SD-LoRA | 35.12 | 0.37 | 0 |
| SD-LoRA-RR | 35.12 | 0.23 | 0 |

suggests that the performance gains of SD-LoRA cannot be attributed solely to decoupling. Instead, the synergistic effect of training multiple decoupled LoRA components is essential for achieving satisfactory results. Last, we fix learned LoRA components without magnitude rescaling, and also observe a noticeable performance decline. This suggests that the rescaling mechanism enables SD-LoRA to navigate low-loss paths by reweighting contributions from earlier components.

**Analysis of Computation, Parameter, and Storage Efficiency.** As presented in Table 5, we compare the inference computations in terms of GFLOPs, trainable parameters, and feature storage requirements of various CL methods. Notably, InfLoRA (Liang & Li, 2024) and the proposed SD-LoRA eliminate the need for task-specific prompt selection during inference, thereby enjoying the highest inference efficiency. Moreover, our proposed SD-LoRA-RR is capable of further reducing the number of LoRA parameters without reliance on sample rehearsal, making it an ideal choice for resource-constrained CL scenarios.

## 5    CONCLUSION AND DISCUSSION

In this paper, we have introduced SD-LoRA, a computational method designed to address scalability challenges in class-incremental learning with foundation models. By decoupling the learning of magnitude and direction of LoRA components, SD-LoRA provides a rehearsal-free, inference-efficient, and end-to-end optimized solution. Our empirical and theoretical analysis demonstrates that SD-LoRA uncovers a low-loss trajectory that converges to an overlapping low-loss region for all learned tasks, effectively balancing stability and plasticity. Extensive experiments confirmed the effectiveness of SD-LoRA in mitigating catastrophic forgetting while maintaining adaptability to new tasks. Additionally, our two parameter-efficient variants, SD-LoRA-RR and SD-LoRA-KD, further enhance its practicality for resource-constrained applications.

While SD-LoRA has shown promise, several avenues for future research warrant exploration. First, extending SD-LoRA to other foundation models beyond ViTs could provide valuable insights into its generality and effectiveness across different backbone architectures. Second, integrating SD-LoRA with other PEFT techniques, such as adapters or prefix-tuning, may further enhance its performance and scalability. Finally, developing more theoretically grounded strategies for rank reduction and knowledge distillation within SD-LoRA could lead to additional improvements in parameter efficiency and overall performance.

ACKNOWLEDGEMENTS

We would like to thank Ziye Ma and Xinyuan Song for helping formalize the proofs, and Xuelin Liu for assistance with the plots and diagrams. This work was supported in part by the National Key R&D Program of China (2020YFA0713900), the Hong Kong RGC General Research Fund (11220224), the CityU Applied Research Grant (9667264), the National Natural Science Foundation of China under the Tianyuan Fund for Mathematics (12426105) and under Grant 62306233, and the Major Key Project of PCL (PCL2024A06).

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

# A APPENDIX

## A.1 PROOF OF THEOREM 1

In this section, we prove Theorem 1 as a specific case of Theorem 2.

**Assumption 1.** The optimal updates $\Delta\mathbf{W}_t^\star, t \in \{1, 2, \ldots, N\}$ lie within a small neighborhood in the loss landscape, meaning there exists a small constant $\epsilon_1 > 0$ such that $\|\Delta\mathbf{W}^\star - \Delta\mathbf{W}_t^\star\|_{\mathrm{op}} < \epsilon_1$.

**Assumption 2.** The first $j + 1$ singular values of $\Delta\mathbf{W}_t^\star$ are distinct, *i.e*, $\sigma_1 > \ldots > \sigma_j > \sigma_{j+1}$.

**Theorem 2.** *Fix any $j \leq r$. Suppose $\sigma_{j+1} < \sigma_j$, choose any $\gamma \in (0, 1)$ such that $\frac{\sigma_{j+1}}{\sigma_j} \leq \gamma$. Pick any stepsize $\eta \leq \min\{\frac{\gamma\sigma_j^2}{600\sigma_1^3}, \frac{(1-\gamma)\sigma_j}{20\sigma_j^2}\}$. For any $c_\rho < 1$, let the initialization size $\rho$ satisfy*

$$\rho \leq \min\left\{\frac{1}{3}, \frac{1-\gamma}{24}, \frac{c_\rho\sigma_1}{12(m+n+r)\sqrt{\frac{1-\gamma}{24}}\sqrt{\sigma_j}}\right\},$$

*and*

$$\rho \leq \min\left\{\left(\frac{(1-\gamma)c_\rho\sigma_j}{1200(m+n+r)j\sigma_1}\right)^{\frac{2(1+\gamma)}{1-\gamma}}, \left(\frac{\gamma\sigma_j^2}{16000r\sigma_j^2}\right)^{\frac{1+\gamma}{1-\gamma}}, \frac{\gamma\sigma_j\sqrt{2j}}{16\sigma_j\sqrt{m+n+r}}\right\}$$

*Define*

$$T_1 = \left\lfloor\frac{\log\left(\frac{12(m+n+r)\sqrt{\frac{1-\gamma}{24}}\sqrt{\sigma_j}}{c_\rho\rho\sqrt{\sigma_j}}\right)}{\log(1 + \frac{1+\gamma}{2}\eta\sigma_j)}\right\rfloor + 1, \qquad T_2 = \left\lfloor\frac{\log\left(\sqrt{\frac{24}{1-\gamma}}\right)}{\log(1 + 0.1\eta\sigma_j)}\right\rfloor + 1,$$

$$T_3 = \left\lfloor\frac{\log(\rho^{\frac{1-\gamma}{2(1+\gamma)}}/3)}{\log(1 - \frac{3}{2}\eta\sigma_j)}\right\rfloor + 1, \qquad T = \left\lfloor\frac{\log(\rho^{\frac{1-\gamma}{2(1+\gamma)}}/\rho)}{\log(1 + \gamma\eta\sigma_j)}\right\rfloor$$

*Define $T_0 := T_1 + T_2 + T_3$, then we have*

$$\frac{T_0}{T} \leq 1 - \frac{(3 - 2\gamma)(1 - \gamma)}{6(3\gamma + 1)}.$$

*Furthermore, there exists a universal constant $C$ such that with probability at least $1 - (Cc_\rho)^{r-j+1} - C\exp(-r/C)$, for all $T_0 \leq i \leq T$, we have*

$$\|\mathbf{A}_i\mathbf{B}_i - \Delta\mathbf{W}^{[:j]}\|_{\mathrm{op}} \leq 8\rho^{\frac{\delta}{2(2-\sigma)}}\sigma_1 + 4\rho^{\frac{\delta}{2(2-\delta)}}\sqrt{2j}\sigma_1 + \epsilon_1. \tag{A.9}$$

*Proof.* The sequential training of LoRA can be conceptualized as a matrix approximation problem (Jiang et al., 2023):

$$\ell(\mathbf{A}, \mathbf{B}) = \frac{1}{2}\|\mathbf{AB} - \Delta\mathbf{W}_t^\star\|_F^2, \tag{A.10}$$

where $\Delta\mathbf{W}_t^\star$ denotes the optimal update matrix for the current $t$-th task. We compute the gradients of $\ell$ with respect to $\mathbf{A}$ and $\mathbf{B}$, respectively:

$$\nabla_\mathbf{A}\ell = (\mathbf{AB} - \Delta\mathbf{W}_t^\star)\mathbf{B}^\mathsf{T}$$

$$\nabla_\mathbf{B}\ell = \mathbf{A}^\mathsf{T}(\mathbf{AB} - \Delta\mathbf{W}_t^\star).$$

Then, the gradient descent updates with a step size of $\eta$ are

$$\mathbf{A}_+ = \mathbf{A} - \eta\nabla_\mathbf{A}\ell = \mathbf{A} + \eta(\Delta\mathbf{W}_t^\star - \mathbf{AB})\mathbf{B}^\mathsf{T},$$

$$\mathbf{B}_+ = \mathbf{B} - \eta\nabla_\mathbf{B}\ell = \mathbf{B} + \eta\mathbf{A}^\mathsf{T}(\mathbf{AB} - \Delta\mathbf{W}_t^\star).$$

By performing the singular value decomposition (SVD) of $\Delta\mathbf{W}_t^\star$, *i.e*,

$$\Delta\mathbf{W}_t^\star = \boldsymbol{\Phi}_t\boldsymbol{\Sigma}_t\boldsymbol{\Psi}_t^\mathsf{T},$$

and exploiting the rotational invariance of the Frobenius norm in Eqn. (A.10), *i.e*, by substituting $\mathbf{A} \to \mathbf{\Phi_t}^\mathsf{T}\mathbf{A}$ and $\mathbf{B} \to \mathbf{\Psi}_t^\mathsf{T}\mathbf{B}$, we may assume without loss of generality that $\Delta\mathbf{W}_t^\star$ is diagonal.

We then rewrite $\Delta\mathbf{W}_t^\star$ as

$$\Delta\mathbf{W}_t^\star = \begin{pmatrix} \mathbf{\Sigma}_t & \mathbf{0} \\ \mathbf{0} & \mathbf{\Sigma}_t' \end{pmatrix}$$

with $\mathbf{\Sigma}_t = \mathrm{diag}(\sigma_1, \ldots, \sigma_j) \in \mathbb{R}^{j \times j}$ and $\mathbf{\Sigma}_t' \in \mathbb{R}^{(m-j)\times(n-j)}$ be a diagonal matrix with $\sigma_{j+1}, \ldots, \sigma_{\min\{m,n\}}$ on the diagonals. We next introduce the following block partitions:

$$\mathbf{A} = \begin{pmatrix} \mathbf{U} \\ \mathbf{J} \end{pmatrix} \quad \text{and} \quad \mathbf{B} = (\mathbf{V} \ \mathbf{K}),$$

where

$$\mathbf{U} \in \mathbb{R}^{j \times r}, \quad \mathbf{J} \in \mathbb{R}^{(m-j)\times r}, \quad \mathbf{V} \in \mathbb{R}^{r \times j}, \quad \mathbf{K} \in \mathbb{R}^{r \times (n-j)}.$$

In this way, we can decompose $\mathbf{AB} - \Delta\mathbf{W}_t^\star$ as

$$\mathbf{AB} - \Delta\mathbf{W}_t^\star = \begin{pmatrix} \mathbf{UV} - \mathbf{\Sigma}_t & \mathbf{UK} \\ \mathbf{JV} & \mathbf{JK} - \mathbf{\Sigma}_t' \end{pmatrix},$$

and we are ready to bound the difference $\mathbf{AB} - \Delta\mathbf{W}^\star$:

$$\begin{aligned}
\|\mathbf{AB} - \Delta\mathbf{W}^\star\|_{\mathrm{op}} &= \|\mathbf{AB} - \Delta\mathbf{W}_t^\star + \Delta\mathbf{W}_t^\star - \Delta\mathbf{W}^\star\|_{\mathrm{op}} \\
&\leq \|\mathbf{AB} - \Delta\mathbf{W}_t^\star\|_{\mathrm{op}} + \|\Delta\mathbf{W}_t^\star - \Delta\mathbf{W}^\star\|_{\mathrm{op}} \\
&\leq \|\mathbf{UV} - \mathbf{\Sigma}_t\|_{\mathrm{op}} + \|\mathbf{UK}\|_{\mathrm{op}} + \|\mathbf{JV}\|_{\mathrm{op}} + \|\mathbf{JK} - \mathbf{\Sigma}_t'\|_{\mathrm{op}} + \epsilon_1,
\end{aligned}$$

where we note that $\mathbf{\Sigma}_t = \Delta\mathbf{W}_t^{[:j]}$ (after diagonalization of $\Delta\mathbf{W}_t^\star$). To ensure convergence, it suffices to show that the dominant term $\mathbf{UV}$ approaches $\mathbf{\Sigma}_t$ while the error terms $(\mathbf{J}, \mathbf{K})$ remain small. Let us first extract the gradient descent update of the top left block $\mathbf{U}$:

$$\begin{aligned}
\mathbf{U}_+ &= \mathbf{U} + \eta\Big[ (\mathbf{\Sigma}_t - \mathbf{UV})\mathbf{V}^\mathsf{T} + (-\mathbf{UK})\mathbf{K}^\mathsf{T} \Big] \\
&= \mathbf{U} + \eta\left( \mathbf{\Sigma}_t\mathbf{V}^\mathsf{T} - \mathbf{U}(\mathbf{VV}^\mathsf{T} + \mathbf{KK}^\mathsf{T}) \right).
\end{aligned}$$

Similarly, we have

$$\mathbf{V}_+ = \mathbf{V} + \eta\Big( \mathbf{U}^\mathsf{T}\mathbf{\Sigma}_t - (\mathbf{U}^\mathsf{T}\mathbf{U} + \mathbf{J}^\mathsf{T}\mathbf{J})\mathbf{V} \Big),$$

$$\mathbf{J}_+ = \mathbf{J} + \eta\Big( \mathbf{\Sigma}_t'\mathbf{K}^\mathsf{T} - \mathbf{J}(\mathbf{VV}^\mathsf{T} + \mathbf{KK}^\mathsf{T}) \Big),$$

$$\mathbf{K}_+ = \mathbf{K} + \eta\Big( \mathbf{J}^\mathsf{T}\mathbf{\Sigma}_t' - (\mathbf{U}^\mathsf{T}\mathbf{U} + \mathbf{J}^\mathsf{T}\mathbf{J})\mathbf{K} \Big).$$

To account for the potential imbalance of $\mathbf{U}$ and $\mathbf{V}$, we introduce the following quantities,

$$\mathbf{F} = \frac{\mathbf{U} + \mathbf{V}^\mathsf{T}}{2} \quad \text{and} \quad \mathbf{G} = \frac{\mathbf{U} - \mathbf{V}^\mathsf{T}}{2},$$

so that

$$\mathbf{U} = \mathbf{F} + \mathbf{G} \quad \text{and} \quad \mathbf{V}^\mathsf{T} = \mathbf{F} - \mathbf{G}.$$

Then, the updates for $\mathbf{F}$ and $\mathbf{G}$ are given by

$$\begin{aligned}
\mathbf{F}_+ &= \tfrac{1}{2}\Big( \mathbf{U}_+ + \mathbf{V}_+^\mathsf{T} \Big) \\
&= \tfrac{1}{2}\Big[ \mathbf{U} + \mathbf{V}^\mathsf{T} + \eta\Big( \mathbf{\Sigma}_t\mathbf{V}^\mathsf{T} + \mathbf{\Sigma}_t\mathbf{U} \Big) - \eta\Big( \mathbf{U}(\mathbf{VV}^\mathsf{T} + \mathbf{KK}^\mathsf{T}) + \mathbf{V}^\mathsf{T}(\mathbf{U}^\mathsf{T}\mathbf{U} + \mathbf{J}^\mathsf{T}\mathbf{J}) \Big) \Big] \\
&= \mathbf{F} + \eta\,\mathbf{\Sigma}_t\mathbf{F} - \tfrac{\eta}{2}\Big[ (\mathbf{F}+\mathbf{G})\Big(\mathbf{VV}^\mathsf{T} + \mathbf{KK}^\mathsf{T}\Big) + (\mathbf{F}-\mathbf{G})\Big(\mathbf{U}^\mathsf{T}\mathbf{U} + \mathbf{J}^\mathsf{T}\mathbf{J}\Big) \Big].
\end{aligned}$$

A similar computation gives

$$\begin{aligned}
\mathbf{G}_+ &= \tfrac{1}{2}\Big( \mathbf{U}_+ - \mathbf{V}_+^\mathsf{T} \Big) \\
&= \mathbf{G} - \eta\mathbf{\Sigma}_t\mathbf{G} - \tfrac{\eta}{2}\Big[ (\mathbf{F}+\mathbf{G})\Big(\mathbf{VV}^\mathsf{T} + \mathbf{KK}^\mathsf{T}\Big) - (\mathbf{F}-\mathbf{G})\Big(\mathbf{U}^\mathsf{T}\mathbf{U} + \mathbf{J}^\mathsf{T}\mathbf{J}\Big) \Big].
\end{aligned}$$

It is now natural to introduce the following equations:

$$\mathbf{P} = \mathbf{\Sigma}_t - \mathbf{FF}^\mathsf{T} + \mathbf{GG}^\mathsf{T}, \qquad \mathbf{Q} = \mathbf{FG}^\mathsf{T} - \mathbf{GF}^\mathsf{T},$$

so that one can verify that

$$\mathbf{P} + \mathbf{Q} = \mathbf{\Sigma}_t - \mathbf{UV}.$$

According to the Proposition B.2 and B.5 of Jiang et al. (2023), it holds with high probability that for any $T_1 + T_2 + T_3 \leq i \leq T$,

$$\|\mathbf{U}_i \mathbf{K}_i\|_{\mathrm{op}} \leq 3\rho^{\frac{1-\gamma}{2(1+\gamma)}} \sigma_1, \quad \|\mathbf{J}_i \mathbf{V}_i\|_{\mathrm{op}} \leq 3\rho^{\frac{1-\gamma}{2(1+\gamma)}} \sigma_1,$$

$$\|\mathbf{J}_i \mathbf{K}_i\|_{\mathrm{op}} \leq \rho^{\frac{1-\gamma}{(1+\gamma)}} \sigma_1, \quad \|\mathbf{Q}_i\|_{\mathrm{op}} \leq 4\rho^{\frac{1-\gamma}{2(1+\gamma)}} \sqrt{2j}\sigma_1.$$

$$\|\mathbf{P}_i\|_{\mathrm{op}} \leq (1 - 0.79\eta\sigma_j)^2 + 6\eta^2\sigma_1^2 \|\mathbf{P}_{i-1}\|_{\mathrm{op}} + 80\eta\rho^{\frac{1-\gamma}{1+\gamma}j} j\sigma_1$$

$$\leq (1 - \frac{3\eta\sigma_j}{2})\|\mathbf{P}_{i-1}\|_{\mathrm{op}} + 80\eta\rho^{\frac{1-\gamma}{1+\gamma}j} j\sigma_1^2$$

$$\leq 2\left(1 - \frac{3\eta\sigma_j}{2}\right)^{i-T_1-T_2} \sigma_1 + \frac{80\rho^{\frac{1-\gamma}{1+\gamma}j} j\sigma_1^2}{\sigma_r}.$$

Given that $T_3 = \left\lfloor \frac{\log(\rho^{\frac{1-\gamma}{2(1+\gamma)}}/3)}{\log(1-\frac{3}{2}\eta\sigma_j)} \right\rfloor + 1$, it follows that for all $i$ satisfying $T_1 + T_2 + T_3 \leq i \leq T$, we have $\|\mathbf{P}_i\|_{\mathrm{op}} \leq \rho^{\frac{1-\gamma}{2(1+\gamma)}} \sigma_1$.

$$\|\mathbf{U}_i \mathbf{V}_i - \mathbf{\Sigma}_t\|_{\mathrm{op}} = \|\mathbf{P}_i + \mathbf{Q}_i\|_{\mathrm{op}} \leq \|\mathbf{P}_i\|_{\mathrm{op}} + \|\mathbf{Q}_i\|_{\mathrm{op}} \leq \rho^{\frac{1-\gamma}{2(1+\gamma)}} \sigma_1 + 4\rho^{\frac{1-\gamma}{2(1+\gamma)}} \sqrt{2j}\sigma_1.$$

By combining these parts, we can have

$$\|\mathbf{A}_i \mathbf{B}_i - \Delta\mathbf{W}^{[:j]}\|_{\mathrm{op}} \leq 8\rho^{\frac{1-\gamma}{2(1+\gamma)}} \sigma_1 + 4\rho^{\frac{1-\gamma}{2(1+\gamma)}} \sqrt{2j}\sigma_1 + \epsilon_1 = (8\rho^{\frac{1-\gamma}{2(1+\gamma)}} + 4\rho^{\frac{1-\gamma}{2(1+\gamma)}} \sqrt{2j})\sigma_1 + \epsilon_1$$

$$\square$$

## A.2 Additional Results for Sec. 3.3

To further investigate the temporal evolution of learned LoRA magnitudes $\mathcal{M} = \{\alpha_k\}_{k=1}^N$ in SD-LoRA, we conduct additional experiments on ImageNet-R (Boschini et al., 2022) with an extended task length of $N = 20$ and a more challenging DomainNet dataset (Peng et al., 2019). As visualized in Fig. 6, both experimental configurations reveal a systematic decrease in $\alpha_k$ values throughout the training process. These results corroborate the descending trend observed in our main experiments, demonstrate the consistent behaviors of SD-LoRA across extended task horizons and diverse domain distributions, and align with the theoretical analysis presented in Sec. 3.4.

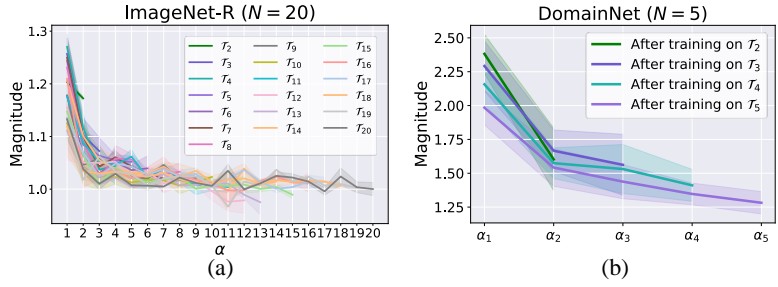

Figure 6: Learned LoRA magnitudes $\mathcal{M} = \{\alpha_k\}_{k=1}^N$ in SD-LoRA on **(a)** ImageNet-R ($N = 20$) and **(b)** DomainNet ($N = 5$).

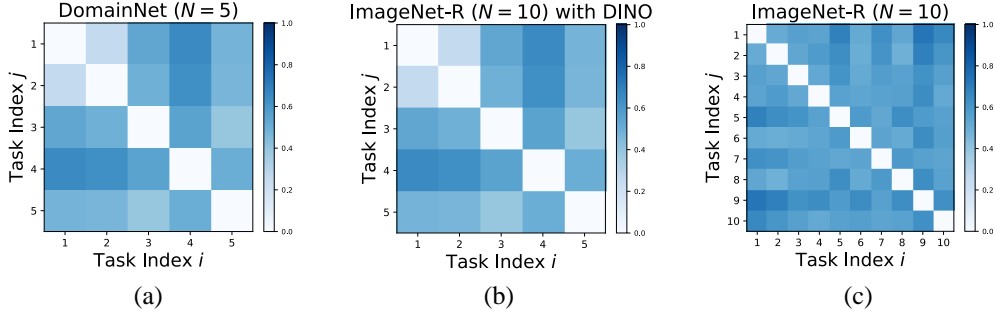

(a)        (b)        (c)

Figure 7: Relative distances computed on **(a)** DomainNet ($N = 5$), **(b)** ImageNet-R ($N = 5$) using ViT-B/16 from DINO, and **(c)** ImageNet-R($N = 10$), respectively.

### A.3 RESULTS ON OTHER CL BENCHMARKS

In addition to ImageNet-R, ImageNet-A, and DomainNet, we evaluate the proposed SD-LoRA on two other widely recognized benchmarks: CIFAR-100 (Krizhevsky, 2009) and CUB-200 (Wah et al., 2011). CIFAR-100 is a standard dataset for image classification, comprising $60,000$ images evenly distributed across 100 classes, with 600 images per class. For our experiments, we split CIFAR-100 into ten tasks, each containing ten classes. Similarly, CUB-200 is a fine-grained dataset specifically designed for bird classification, which consists of $11,788$ images across 200 classes. We divide this dataset into ten tasks, with each task encompassing 20 species. As shown in Table 6, the proposed SD-LoRAs consistently deliver outstanding performance on both datasets.

Additionally, we provide supplementary results to those in Fig. 2(a) by analyzing the relative distances between fine-tuned and pre-trained weights across different benchmarks, backbones, and task lengths. As shown in Fig. 7, the observed trends remain consistent with Finding 1.

Table 6: Performance comparison on CIFAR100 and CUB200.

| Method | CIFAR100 | | CUB200 | |
| --- | --- | --- | --- | --- |
| | Acc ↑ | AAA ↑ | Acc ↑ | AAA ↑ |
| Full Fine-Tuning | $69.49_{(0.50)}$ | $80.35_{(0.87)}$ | $51.43_{(1.41)}$ | $69.74_{(0.93)}$ |
| L2P (Wang et al., 2022b) | $83.18_{(1.20)}$ | $87.69_{(1.05)}$ | $65.18_{(2.49)}$ | $76.12_{(1.27)}$ |
| DualPrompt (Wang et al., 2022a) | $81.48_{(0.86)}$ | $86.41_{(0.66)}$ | $68.00_{(1.06)}$ | $79.40_{(0.88)}$ |
| CODA-Prompt (Smith et al., 2023) | $86.31_{(0.12)}$ | $90.67_{(0.22)}$ | $71.92_{(0.33)}$ | $78.76_{(0.65)}$ |
| InfLoRA (Liang & Li, 2024) | $86.75_{(0.35)}$ | $91.72_{(0.15)}$ | $70.82_{(0.23)}$ | $81.39_{(0.14)}$ |
| SD-LoRA | $\mathbf{88.01}_{(0.31)}$ | $\mathbf{92.54}_{(0.18)}$ | $\mathbf{77.48}_{(0.20)}$ | $\mathbf{85.59}_{(0.44)}$ |
| SD-LoRA-RR | $87.26_{(0.22)}$ | $92.05_{(0.31)}$ | $76.35_{(0.28)}$ | $83.89_{(0.35)}$ |
| SD-LoRA-KD | $87.09_{(0.45)}$ | $92.01_{(0.33)}$ | $75.95_{(0.55)}$ | $83.21_{(0.31)}$ |

