# OpenReview forum: "SD-LoRA: Scalable Decoupled Low-Rank Adaptation for Class Incremental Learning"
_ICLR.cc/2025/Conference — ICLR 2025 Oral_

### Official Review · Reviewer_Uver · 2024-10-25

**Soundness:** 3
**Presentation:** 3
**Contribution:** 3
**Rating:** 8
**Confidence:** 4

**Summary:**

This manuscript focused on continual learning with the pre-trained model (CL-PTM). By indicating that the existing prompt-based methods rely on an unreliable prompt selection mechanism which can lead to the scalability issue, this manuscript proposed Scalable Low-Rank Adaptation (S-LoRA). Specifically, S-LoRA incrementally decouples the learning of the direction and magnitude of Low-Rank Adaptation (LoRA) parameters. The theoretical and empirical analysis indicated that the proposed method tends to follow a low-loss trajectory that converges to an overlapped low-loss region. Experiments on standard benchmarks were conducted to support the proposed method.

**Strengths:**

1. The idea of decoupling the learning of LoRA’s direction and magnitude is novel to me. This method addresses issues with scalability and efficiency in class-incremental learning, providing a valuable contribution to the field.
2. The motivation of this manuscript was driven by extensive empirical support. Some interesting findings were observed through the experiments, which were not investigated before.
3. Theoretical analyses were conducted to support the empirical findings. The explanation of the shared low-loss region and how S-LoRA converges to it effectively supports the results presented.

**Weaknesses:**

1. Some core ideas were not explained clearly. The description of how the decoupling of magnitude and direction need to be better defined.
2. While S-LoRA shows strong performance, some newer methods outside the scope of the prompt-based methods need to be compared for further validation.

**Questions:**

1. Regarding the magnitude and the direction of the LoRA parameters. In Section 4.1, the decomposition of magnitude and direction of LoRA's parameter updates $\Delta \mathbf{W}$ was only casually mentioned. However, I wonder how the magnitude and direction of such a matrix were defined in the authors' derivation. I believe they should be clearly defined to help the readers to understand the core idea of your method.
2. In Findings 3, the concept of "low-loss path", "linear low-loss path", and "low-loss region" need to be explained with more intuitive expressions.
3. In the first equation after Algorithm 1, I wonder if there is a typo for the definition of $r$. Should the first term in RHS be $\mathbf{A}_j \mathbf{B}_j$ rather than $\mathbf{A}_i \mathbf{B}_i$? If not, please provide further explanations about this point.
4. Still with the same equation. Since the RHS consists of a linear combination of "directions" mentioned by the authors, I wonder how "the residual" of a direction approximation should be defined. In my understanding, $r$ should also be a direction analog to the operations on vectors. However, the authors also had some expressions like "the residual $r$ is less than the threshold $\tau$". I didn't get how to control a residual direction by a scalar threshold.
5. Before Theorem 1 in Section 5, the authors mentioned that $\|\cdot\|$ refers to the operator norm. Do all $\|\cdot\|$ notations within the manuscript stand for the operator norm?
6. I noticed that the proposed S-LoRA method was mainly compared to the prompt-based baselines except for InfoLoRA. However, S-LoRA is not an improvement of the prompt-based methods. It would be better if the authors provide comparisons with a wide range of recent studies like EASE [1], or at least some recent prompt-based methods like [2-5].

I will accordingly change my rating if my concerns are addressed.

References:
[1] Expandable Subspace Ensemble for Pre-Trained Model-Based Class-Incremental Learning. CVPR 2024.
[2] RCS-prompt: learning prompt to rearrange class space for prompt-based continual learning. ECCV 2024.
[3] Prompt Gradient Projection For Continual Learning. ICLR 2024.
[4] PromptFusion: Decoupling Stability and Plasticity for Continual Learning. ECCV 2024.
[5] Consistent Prompting for Rehearsal-Free Continual Learning. CVPR 2024.

---

> ### Author Response · Authors · 2024-11-21
> **To Reviewer Uver (Part I)**
>
> We appreciate very much your constructive comments on our paper. Please kindly find our response to your comments below, and all revisions made to the paper are highlighted in blue for your ease of reference. We hope that our response satisfactorily addresses the issues you raised. Please feel free to let us know if you have any additional concerns or questions.
>
> **Q1**: Better clarification about the magnitude and direction of the LoRA parameters.
> > - In the vanilla LoRA framework, the weight update is expressed as $\Delta W = AB = \\|AB\\|_F \cdot \frac{AB}{\\|AB\\|_F} = \\|AB\\|_F \cdot \overline{AB}$. This decomposition highlights two components: $\\|AB\\|_F$, the Frobenius norm of $AB$, which represents the learned magnitude, and $\overline{AB} = \frac{AB}{\\|AB\\|_F}$, the normalized matrix indicating the direction.
> > - In S-LoRA,  we preserve only the directions learned from previously completed tasks (i.e., $\overline{A_iB_i}$ for $i = 1, \dots, j-1$, where $\mathcal{T}_j$ is the current task). Their associated weights $\alpha_i$ ($i = 1, \dots, j-1$) are treated as learnable parameters. By decoupling magnitude and direction, we empirically demonstrate its effectiveness in improving average performance, as shown in the experiments.
> > - Section 4.2 provides further exploration of how this decoupling helps alleviate forgetting. Specifically, Finding 3 illustrates that the $\Delta W$ learned by S-LoRA better aligns with $\Delta W^*$, which represents the weights located in the shared low-loss region (i.e., the region in the parameter space where the model consistently achieves low loss across all tasks).
> > - A more detailed theoretical explanation is provided in Theorem 1, which shows that the previously learned directions (e.g., $\overline{A_iB_i}$ for $i = 1, 2$) correspond to the principal components of $\Delta W^*$. This also explains why the automatically learned weights (e.g., $\alpha_i$ values for $i = 1, 2$) are larger for earlier tasks compared to those learned for later tasks.
>
> **Q2**: In Findings 3, the concept of "low-loss path", "linear low-loss path", and "low-loss region" need to be explained with more intuitive expressions.
> > - (**Intuitive expression.**) For **Low-Loss Region**: This is a broader area in the parameter space where the model's loss is consistently low. Intuitively, it's like the floor of a valley or plateau where any position results in a good model performance. It represents the parameter configurations that achieve low error across different tasks. **Low-Loss Path:** This refers to a trajectory in the parameter space along which the model's loss remains low. Intuitively, imagine navigating through a landscape of mountains and valleys, where the valleys represent areas of low loss. **Linear Low-Loss Path:** This is a specific type of low-loss path where the trajectory is a straight line in the parameter space. Imagine drawing a straight line across a flat valley; the line stays within the low-loss region, meaning the model's performance remains stable along this direct route.
> >  - Additionally, previous works [1, 2] have shown that the linear low-loss path exists within the context of CL, indicating that the trajectory along this path is effective for maintaining low loss across sequential tasks. In Sec. 4.2, we demonstrate that the proposed S-LoRA can effectively seek a low-loss path to identify the shared low-loss region, thereby achieving improved performance.
> >
> > [1] Mirzadeh, Seyed Iman, et al. "Linear Mode Connectivity in Multitask and Continual Learning." ICLR, 2021.\
> > [2] Verwimp, Eli, Matthias De Lange, and Tinne Tuytelaars. "Rehearsal revealed: The limits and merits of revisiting samples in continual learning." ICCV, 2021.
> >
> **Q3/4**: The typo in the equation after Algorithm 1 (the first term should be $A_jB_j$). How should the 'residual' of this directional approximation be defined, and how can a residual direction be controlled by a scalar threshold?
> > - Thanks for your careful review. We appreciate your attention to detail. This was a typo, and we have already corrected it in the main text, with the changes highlighted in blue.
> > - The updated equation is $r=\\|\\overline{A_jB_j}-\\sum_{i=1}^{j-1}\\hat{\\alpha}_i \\overline{A_iB_i}\\|$
>
> **Q5**: Do all $\\|\cdot\\|$ notaion within the manuscript stand for the operator norm?
> > - Thank you for your comment. The notation $\\|\cdot\\|$ used throughout the manuscript, except in Sec. 5, refers to the Frobenius norm. To avoid confusion, we have revised the manuscript accordingly. Thank you again for helping us improve the clarity of the paper.

---

> ### Author Response · Authors · 2024-11-21
> **To Reviewer Uver (Part II)**
>
> **Q6**: To compare with other methods like EASE[1], or at least some recent prompt-based methods.
> > - Following your suggestion, we have added experiments to compare with EASE [1] as well as recent prompt-based methods[2-4]. Additionally, we have included other LoRA-based methods [5-6] for a more comprehensive comparison.
> > - The results show that, compared to both recent prompt-based and LoRA-based methods, the proposed S-LoRA consistently achieves more competitive performance. This further demonstrates the effectiveness of the proposed S-LoRA.
> >
> > | Methods| IN-R(N=5)(Acc/AAA)|IN-R(N=10)(Acc/AAA)|IN-R(N=20)(Acc/AAA)|
> >|:-:|:-:|:-:|:-:|
> >|EASE[1] | 77.35/81.46 |76.17/81.73 | 70.58/78.31 |
> >|PGP[2] | 71.00/75.04 |69.50/74.14 | 66.94/77.98 |
> >|ADAM[3] | 74.27/79.99 |72.87/79.39 | 70.47/77.29 |
> >|RanPAC[4] | 76.95/82.79 | 74.58/81.77 | 72.40/79.27 |
> >|O-LoRA[5] | 73.88/78.89  | 70.65/76.43   |  65.23/71.89|
> >|Task Arithmetic LoRA[6]| 72.77/78.07 | 71.02/76.89     | 63.29/70.88    |
> >|InfLoRA[7]| 76.95/81.81| 74.75/80.67 | 69.89/76.68 |
> >|S-LoRA|**79.15**/**83.01**  | **77.34**/**82.04** | **75.26**/**80.22**|
> >
> > [1] Zhou, Da-Wei, et al. "Expandable subspace ensemble for pre-trained model-based class-incremental learning." CVPR, 2024.\
> > [2] Qiao, Jingyang, et al. "Prompt Gradient Projection for Continual Learning." ICLR, 2024.\
> > [3] Zhou, Da-Wei, et al. "Revisiting class-incremental learning with pre-trained models: Generalizability and adaptivity are all you need." IJCV, 2024.\
> > [4] McDonnell, Mark D., et al. "Ranpac: Random projections and pre-trained models for continual learning." NeurIPS, 2024. \
> > [5] Wang, Xiao, et al. "Orthogonal Subspace Learning for Language Model Continual Learning." EMNLP, 2023.\
> > [6] Chitale, Rajas, et al. "Task Arithmetic with LoRA for Continual Learning." arXiv preprint arXiv:2311.02428 (2023).\
> > [7] Liang, Yan-Shuo, and Wu-Jun Li. "InfLoRA: Interference-Free Low-Rank Adaptation for Continual Learning." CVPR, 2024.

---

> ### Comment · Reviewer_Uver · 2024-11-23
> **Comments after Author Rebuttal**
>
> Thanks for responding to my questions. I carefully read them and my previous concerns have been addressed. The manuscript became much more readable after the clarifications and corrections of some core points, e.g., clarifying the precise definition of "residual" and making it compatible with your definitions of "magnitude" and "direction", eliminating the confusion regarding different norms, etc. Furthermore, the authors provided additional comparisons with diverse baselines.
>
> Based on the author's response, I decided to increase my rating to 8. Hope the additional results can be included in the final version of this manuscript.

---

### Official Review · Reviewer_on1i · 2024-10-29

**Soundness:** 3
**Presentation:** 4
**Contribution:** 4
**Rating:** 8
**Confidence:** 5

**Summary:**

This paper presents a new method called S-LoRA for addressing the Class Incremental Learning (CIL) problem. Currently, prompt-based methods are the most effective method that does not rely on memory-buffer methods in CIL. However, as highlighted by the authors, these approaches have some limitations, particularly regarding prompt selection: they could be more efficient since they require double passes through the model to identify similar prompts and depend heavily on choosing the correct prompt. The proposed method (S-LoRA) aims to alleviate these issues by enabling each task to learn a set of LoRA weights over a pre-trained model, along with a separate set of coefficients that determine the importance of each LoRA-weight. The authors provide theoretical and empirical evidence demonstrating that their proposed method achieves strong results across multiple benchmarks.

**Strengths:**

- The paper is very well written. It presents the constraints and motivations the authors want to work with and clearly explains a simple but effective method. The experiments shown support the majority of what is presented in the paper.
- The S-LoRA method represents an advancement over current methods that use pre-trained models. It mitigates the issues introduced by prompt-based approaches and reduces the inference time. The idea of splitting the LoRA trainable weights into direction and magnitude is interesting, as it helps each task have the plasticity to add new information while encouraging the reuse of weights learned in past tasks.
    - The authors mentioned that an ideal CL method needs to be rehearsal-free, which I may partially agree with as this depends strongly on the context and scenario of the problem. However, I do agree that it needs to be more efficient at the time of inference.
- The experiments performed in Section 4 help to understand the proposed method's behaviour. It is also interesting to find limitations in the model and propose alternatives that mitigate these problems.
    - Moreover, they are well accompanied by a suitable experimental ablation.

**Weaknesses:**

- Some sections of the paper mention that the proposed method improves up to 31.05% across various CL benchmarks. It could be helpful to change this as it can present confusion, especially as this number does not necessarily relate to the current best method.
- Algorithm 1 needs to be more precise. In lines 1 and 2, you assume you are working in a Task_j, but then iterate over the T task in lines 6 to 8. This iteration may confuse readers and make them think that a task identifier conditions parameters alpha, A_j and B_j.

**Questions:**

- I agree with the first finding shown in section 4.2. However, this can change drastically when the input data distribution changes abruptly.
    - I understand that this may be outside the scope of the paper, but did you measure the relative distance with other benchmarks? It would be helpful to compare elements outside the ViT pre-training distribution.
- Can you explain the intuition behind the lack of forgetting in the proposed method?
    - For example, Figure 3.A shows how the alpha values change as different tasks are trained. I imagine this may strongly affect the representation generated for the first task examples, which are only expected to be affected by the weights of A_1 and B_1.
    - Do you have forgetting results?
- Some implementation details are not clearly explained:
    - How are alphas initialised? In most routing methods, the initialisation of the alphas plays a critical role; I can imagine that this can also happen here.
    - Does any regularisation apply? Or do you train them freely?
- Can you explain how the values in Table 5 were calculated?
    - Specifically, I have doubts about the comparison of L2P's trainable parameters versus ES-LoRA's. In L2P, only the prompts pool is trained, which should be smaller than LoRA's A and B.
    - On the other hand, in ES-LoRA, the A and B values are accumulated, which is not accounted for in the table.
    - Can you add the S-LoRA values for comparison?

---

> ### Author Response · Authors · 2024-11-21
> **To Reviewer on1i  (Part I)**
>
> Thank you sincerely for your thoughtful and positive feedback on our work. We are particularly grateful for your recognition of the various aspects of our research. Below, we have provided a detailed explanation for your remaining concern as follows. Please do not hesitate to let us know if you have any further questions.
>
> **Q1**: I understand that this may be outside the scope of the paper, but did you measure the relative distance with other benchmarks? It would be helpful to compare elements outside the ViT pre-training distribution.
> > - To address your question, we have conducted additional experiments and plotted the relative distances across different benchmarks and backbones, including DomainNet and the DINO backbone.
> > - Since ImageNet-R is an out-of-distribution task relative to the ViT pre-training distribution [1], we further extended the original paper's results on ImageNet-R (N=5) to ImageNet-R (N=10) for a more comprehensive comparison with tasks outside the ViT pre-training distribution.
> > - **Please refer to the Appendix A.2 for reference.** The results demonstrate that the same trend, consistent with Finding 1, is observed across various benchmarks, backbones, and task lengths.
> >
> >[1] Hendrycks, Dan, et al. "The many faces of robustness: A critical analysis of out-of-distribution generalization." CVPR, 2021.
>
> **Q2**: The intuition behind how the proposed method alleviates forgetting, along with the corresponding forgetting results.
> > - Thank you for pointing this out. The explanation of how S-LoRA alleviates forgetting is central to the method and is detailed in Finding 3.
> > - To address your concern about strong impact on performance on $\mathcal{T}_1$, take ImageNet-R(N=5) for example, we have listed the forgetting of $\mathcal{T}_1$, during the incremental training, in the following table. It can be seen that the forgetting on $\mathcal{T}_1$ is smaller than others. This is because, although the $\alpha_1$ changes, the final weights still relatively lie in the low-loss region for $\mathcal{T}_1$ (see Finging 3$(c)$).
> >
> >|IN-R(N=5)|Coda-Prompt|Hide-Prompt|InfLoRA|S-LoRA|
> >|:-:|:-:|:-:|:-:|:-:|
> >|FM($\downarrow$) of $\mathcal{T}_1$ | 7.61 |  7.98 | 9.18| **6.98** |
> >
> > - To further clarify why S-LoRA effectively mitigates forgetting, we provide a detailed explanation below. As demonstrated in Finding 3(a), when the model is trained on the second task $\mathcal{T}_2$, S-LoRA learns the weights ${\alpha_1, \alpha_2}$ and LoRA $\overline{A_2B_2}$ while keeping $\overline{A_1B_1}$ fixed. During training, **S-LoRA first focuses its updates along the critical directions learned from earlier tasks** (i.e., larger $\alpha_1$ on $\overline{A_1B_1}$), enabling the model to quickly approach the shared low-loss region. Then, by incrementally introducing LoRA (i.e., $\overline{A_2B_2}$),  **it fine-tunes the update directions, allowing the model to effectively converge on the shared low-loss region** across all tasks. Specifically,
> >   - By analyzing interpolating points along the update path, we observe that S-LoRA can improve performance on $\mathcal{T}_2$ without degrading performance on $\mathcal{T}_1$, as shown in Finding 3$(c)$. **This result verifies that the weights converged by S-LoRA lie within the shared low-loss region for both tasks.**
> >   - **We theoretically prove that the gradually learned $\\overline{A_iB_i}$ sequentially approximate the principal components of the optimal $\\Delta W^{\ast}$** ($\\Delta W^*= W^*-W_0$, where $W^*$ lies in the shared low-loss region for all tasks.) This, in turn, explains why the model assigns larger weights to the first learned $\overline{A_iB_i}$ components, as these represent the principal directions necessary to achieve the optimal $\\Delta W^*$.
> >  - **In summary**, through our designed S-LoRA, the model can find a low-loss path[1,2] for updates and converge to weights within a common shared low-loss region. This approach effectively mitigates forgetting, **providing a new perspective based on low-loss path** for leveraging LoRA to address this issue.
> >  - For your reference, the table below presents the overall forgetting results (FM), showing that the proposed S-LoRA consistently outperforms other methods in minimizing forgetting.
> >
> >| Methods| IN-R(N=5)(FM$\downarrow$)|IN-R(N=10)(FM$\downarrow$)|IN-R(N=20)(FM$\downarrow$)|
> >|:-:|:-:|:-:|:-:|
> >|Finetune| 28.42 | 30.87  | 39.60  |
> >|L2P| 5.54  | 5.54 | 5.95 |
> >|DualPrompt | 4.63 | 5.58 |  6.22    |
> >|CodaPrompt | 4.03 | 4.87 |  4.45    |
> >|InfLoRA| 4.73 | 5.66 | 7.47 |
> >|HidePrompt| 4.31 | 5.52 | 4.76 |
> >|S-LoRA| **3.98** | **4.32**  | **4.39**   |
> >

---

> ### Author Response · Authors · 2024-11-21
> **To Reviewer on1i (Part II)**
>
> >
> > [1] Mirzadeh, Seyed Iman, et al. "Linear Mode Connectivity in Multitask and Continual Learning." ICLR, 2021.\
> > [2] Verwimp, Eli, Matthias De Lange, and Tinne Tuytelaars. "Rehearsal revealed: The limits and merits of revisiting samples in continual learning." ICCV, 2021.\
> > [3] Liang, Yan-Shuo, and Wu-Jun Li. "InfLoRA: Interference-Free Low-Rank Adaptation for Continual Learning." CVPR, 2024.\
> > [4] Wang, Shipeng, et al. "Training networks in null space of feature covariance for continual learning." CVPR, 2021.
>
> **Q3**: Some implementation details need to be clearly explained like 1) How are alphas initialised? 2) Does any regularisation apply? Or do you train them freely?
> > - At the beginning of a new task, we initialize all $\alpha$ values to 1. To address your concern, we also tested initializing them to 0.1, 0.5 and 0.8. The corresponding results are presented in the following table. As observed, **the initialization of $\alpha$ has a relatively small impact on performance overall, but its influence becomes more noticeable when the task sequence length is large**.
> >
> >| Initialization| IN-R(N=5)(Acc/AAA)|IN-R(N=10)(Acc/AAA)|IN-R(N=20)(Acc/AAA)|
> >|:-:|:-:|:-:|:-:|
> >|$\alpha=(0.1,...,0.1)$| 78.47/82.20 | 76.82/80.59   | 71.72/77.82 |
> >|$\alpha=(0.5,...,0.5)$| 78.63/82.66 | 77.01/81.67  | 74.43/79.89  |
> >|$\alpha=(0.8,...,0.8)$ | 79.02/82.99  | 77.17/81.89  | 74.27/79.95    |
> >|$\alpha=(1,...,1)$| 79.15/83.01  | 77.34/82.04  | 75.26/80.22     |
> >
> > - In the proposed S-LoRA, we **do not apply any additional regularization** when training $\alpha$; they are trained freely.
> >   - In Theorem 1, we theoretically analyzed why the model tends to learn larger weights for previously learned $\overline{A_iB_i}$, as shown in Finding 2.
> >   - The analysis shows that, during sequential training, as $A_iB_i$ gradually approximate the principal components of $\Delta W^*$ (i.e., the weights lie in the shared low-loss region across all tasks), the previously learned $A_iB_i$ naturally take on a more significant role in the model.
>
> **Q4**: Can you explain how the values in Table 5 were calculated?
> > - Let $d$ denote the embedding dimenstion, $e$ denote the prompt length, $p$ refer to the number of prompts, and $l$ is the number of layers in which prompts are inserted. The term "trainable parameters" refers to the amount of parameters that requires gradient preservation and backpropagation during training.
> > - In the training process of L2P, samples in a batch will select different prompts from the entire prompt pool, thus the size of trainable parameters is equivalent to the entire prompt pool, namely $dlp(e+1)$ where $d=768, l=1, e=20, p=30$. In contrast, Dual-Prompt only needs to preserve gradients and backpropagate for global prompts and task-specific prompts of the current task during training, resulting in fewer trainable parameters, namely $de_gl_g+d(e_t+1)l_t$, where $d=768, e_g=6, l_g=2, e_t=20, l_t=3$. The trainable parameters of S-LoRA and ES-LoRA include task-specific LoRAs and negligible $\alpha$. Since it is not necessary to preserve gradients for all LoRAs, the trainable parameters of S-LoRA and ES-LoRA are fewer than those of L2P, namely $4ldr$ for S-LoRA and $4ld\bar{r}$ for ES-LoRA, where $l=12, d=768, r=10, \bar{r}=6.3$.
> > - In Table 5, we primarily focus on the training and inference efficiency. Hence, we present the trainable parameters and FLOPs. The accumulated parameters of $A$ and $B$ in S-LoRA are the same as those in other LoRA-based methods, such as O-LoRA [1]. However, in this paper, we theoretically and empirically demonstrate that the later-learned LoRA components are not as crucial. Therefore, we propose ES-LoRA, which further alleviates this issue and enhances practical efficiency. The complete Table 5, including S-LoRA, is shown in the revised paper.
> >
> > [1] Wang, Xiao, et al. "Orthogonal Subspace Learning for Language Model Continual Learning." EMNLP, 2023.
>
> **W1/2:** 1)Some sections of the paper mention that the proposed method improves up to 31.05% across various CL benchmarks. It could be helpful to change this as it can present confusion. 2) Algorithm 1 needs to be more precise.
> > Thank you for the suggestion to improve readability. We have updated the corresponding section and enhanced the clarity and precision of Algorithm 1 in the revised paper

---

> > ### Comment · Reviewer_on1i · 2024-11-22
> >
> > I thank the authors for the clear response.
> > After reading the comments of the other reviewers and the answers provided by the authors, I decided to raise my score.

---

### Official Review · Reviewer_yo9h · 2024-10-30

**Soundness:** 3
**Presentation:** 4
**Contribution:** 2
**Rating:** 6
**Confidence:** 4

**Summary:**

Existing methods depend on the precision of the selection mechanism. This paper propose a class incremental learning (CLS) method called S-LoRA to decouple the learning of the direction and magnitude of LoRA parameters. It can incrementally learn task-specific LoRA and importance parameters while preserving the optimization directions of previously tasks. The experimental results show their superiority on multiple benchmarks.

**Strengths:**

1. The ablation study is interesting and validate the effectiveness of the learned importance factors $a$.
2. The paper is well organization and the proposed approach is easy followed.

**Weaknesses:**

1. My main concern is that the idea of this paper is just like a MOE-LoRA, which has been studied in [1-2] in continual learning. The learned importance parameters are just like the gates of MOE.  We expect the author to discuss more differences between the proposed "reweighting LoRA" and "MOE LoRA".
2. The experiments are insufficient. There are some other methods in CLS that should be compared, like RanPAC[3], and ADAM[4]. To the best of our knowledge, the RanPAC currently performs best among CIL methods, where the details can be found in this survey[8] about PTM-based CL.  Additionally, some SOTA LoRA methods for continual learning (e.g., O-LoRA[6] and N-LoRA[7]) should be also compared. These two works are very related to LoRA for continual learning, even though they are performed on language process tasks.
3. The two efficient versions of S-LoRA seem unrelated to the core idea of this paper, and the effect is not significant. The author should discuss more details about the ES-LoRA1 and ES-LoRA2.

If the author can answer my issues well, I am willing to improve my score.


Reference

[1] Liu J, Wu J, Liu J, et al. Learning Attentional Mixture of LoRAs for Language Model Continual Learning[J]. arXiv preprint arXiv:2409.19611, 2024.

[2] Dou S, Zhou E, Liu Y, et al. LoRAMoE: Alleviating World Knowledge Forgetting in Large Language Models via MoE-Style Plugin[C]//Proceedings of the 62nd Annual Meeting of the Association for Computational Linguistics (Volume 1: Long Papers). 2024: 1932-1945.

[3] McDonnell, M. D.; Gong, D.; Parvaneh, A.; Abbasnejad, E.; and van den Hengel, A. 2024. Ranpac: Random projections and pre-trained models for continual learning. Advances in Neural Information Processing Systems, 36.

[4] Zhou, D.-W.; Ye, H.-J.; Zhan, D.-C.; and Liu, Z. 2023b. Revisiting class-incremental learning with pre-trained models: Generalizability and adaptivity are all you need. arXiv preprint arXiv:2303.07338.

[6] Xiao Wang, Tianze Chen, Qiming Ge, Han Xia, Rong Bao, Rui Zheng, Qi Zhang, Tao Gui, and Xuanjing Huang. 2023a. Orthogonal subspace learning for language model continual learning. arXiv preprint arXiv:2310.14152.

[7] Yang S, Ning K P, Liu Y Y, et al. Is Parameter Collision Hindering Continual Learning in LLMs?[J]. arXiv preprint arXiv:2410.10179, 2024.

[8] Continual Learning with Pre-Trained Models: A Survey.

**Questions:**

See the Weaknesses.

**Details Of Ethics Concerns:**

1. My main concern is that the idea of this paper is just like a MOE-LoRA, which has been studied in [1-2] in continual learning. The learned importance parameters are just like the gate of MOE.
2. The experiments are insufficient. There are some other methods in CLS that should be compared, like RanPAC[3], and ADAM[4]. To the best of our knowledge, the RanPAC currently performs best among CIL methods, where the details can be found in this survey[8] about PTM-based CL.  Additionally, some SOTA LoRA methods for continual learning (e.g., O-LoRA[6] and N-LoRA[7]) should be also compared. These two works are very related to LoRA for continual learning, even though they are performed on language process tasks.

---

> ### Author Response · Authors · 2024-11-22
> **To Reviewer yo9h (Part I)**
>
> We appreciate very much your constructive comments on our paper. Please kindly find our response to your comments below, and all revisions made to the paper are highlighted in blue for your ease of reference. We hope that our response satisfactorily addresses the issues you raised. Please feel free to let us know if you have any additional concerns or questions.
>
> **W1 & Q1**: Main Concern: the differences between the proposed S-LoRA and MoE-LoRA [1,2]. We expect the author to discuss more differences.
> > Thank you for bringing these papers to our attention. Below, we would like to clarify the key differences between our proposed S-LoRA and MOE-LoRA from the following perspectives.
> >   - **Training Stage**.
> >    MoE-LoRA [1,2] can be viewed as the LoRA version of CODA-Prompt [3], as both rely on a gating mechanism to select prompts (in CODA-Prompt) or LoRA components (in MoE-LoRA), sharing similar properties as outlined in Table 1.
> >     - (***Task-level v.s. Example-level***) The gating mechanism in MoE-LoRA is sample-dependent, where the gating is trained at the example level, meaning **different samples are assigned different weights of LoRAs**. In contrast, S-LoRA uses task-level training, **where all task samples share the same learned $\alpha$**. While sample-dependent gating can sometimes improve accuracy,  as discussed in Lines 78-80, it also creates a bottleneck, since wrong expert selection can significantly degrade performance, especially when samples from different tasks are similar.
> >     - (***Efficiency***) In **MoE-LoRA, all LoRA components/experts are treated equally** and are continually added, leading to inefficiency as the number of tasks increases. S-LoRA, on the other hand, addresses this **through both theoretical and empirical insights, showing that later-learned LoRA components contribute less**. By reducing their ranks, S-LoRA improves efficiency without sacrificing performance.
> > - **Inference Stage**
> >     - (***Fixed v.s. Gating***) During inference, S-LoRA directly evaluates the current model, i.e., the one after learning of task $j$ via ${\bf{W}}_0+\alpha_1\overline{A_1 B_1}+\alpha_2\overline{A_2 B_2}+\cdots+\alpha_j\overline{A_j B_j}$, on previous tasks, where all $\alpha_j$ for $j\in\{1,\cdots,N-1\}$ are fixed thanks to the low-loss path. However, MoE-LoRA requires re-computing weights for each individual sample of a previous task, often resulting in inconsistencies with the optimal weights learned during training. This inconsistency can exacerbate forgetting.
> >     - (***Efficiency***) S-LoRA eliminates additional computational costs associated with MOE-LoRA's dynamic gating. By combining weighted LoRAs back into the foundation model, S-LoRA matches the computational efficiency of the foundation model itself while maintaining strong performance.
> >
> > To better address your concern, we have also re-implemented MoE-LoRA[1] on ImageNet-R and compared it with the proposed S-LoRA for your reference.
> >
> >| Methods| IN-R(N=5)(Acc/AAA)|IN-R(N=10)(Acc/AAA)|IN-R(N=20)(Acc/AAA)|
> >|:-:|:-:|:-:|:-:|
> >|MoE-LoRA[1]|  74.08/81.07   |  70.92/77.81   |   62.97/70.44   |
> >|S-LoRA|**79.15**/**83.01**  | **77.34**/**82.04** | **75.26**/**80.22**|
> >
> >[1] Liu J, Wu J, Liu J, et al. Learning Attentional Mixture of LoRAs for Language Model Continual Learning. arXiv, Sep.29, 2024.\
> >[2] Dou S, Zhou E, Liu Y, et al. LoRAMoE: Alleviating World Knowledge Forgetting in Large Language Models via MoE-Style Plugin. ACL 2024.\
> >[3] Smith, James Seale, et al. "Coda-prompt: Continual decomposed attention-based prompting for rehearsal-free continual learning." CVPR, 2023.

---

> ### Author Response · Authors · 2024-11-22
> **To Reviewer yo9h (Part II)**
>
> **W2 & Q2**: This paper should include a comparison with methods like RanPAC[1], ADAM[2], and especially LoRA-based CL methods such as O-LoRA[3] and N-LoRA[4], even though these are applied to language processing tasks.
> > - Thanks for the suggestion. Since N-LoRA is a concurrent work released on arXiv on Oct. 14, after our paper submission, and no official code is available for it, we have, following your advice, included a comparison with other mentioned methods such as RanPAC[1], ADAM[2], and O-LoRA[3]. The performance results for these methods are presented in the table below.
> > - We can observe that, compared to the latest SOTA methods and closely related LoRA-based approaches, S-LoRA demonstrates competitive performance, further validating the effectiveness of the proposed approach. It is worth noting that O-LoRA [2] applies orthogonal regularization to parameters rather than the feature space [4], which limits its effectiveness in mitigating forgetting. In contrast, the proposed S-LoRA introduces a novel approach by exploring low-loss paths to prevent forgetting, achieving superior performance in the process.
> >
> >| Methods| IN-R(N=5)(Acc/AAA)|IN-R(N=10)(Acc/AAA)|IN-R(N=20)(Acc/AAA)|
> >|:-:|:-:|:-:|:-:|
> >|ADAM[2]| 74.27/79.99 |72.87/79.39 | 70.47/77.29 |
> >|RanPAC[1]| 76.95/82.79 | 74.58/81.77 | 72.40/79.27 |
> >|O-LoRA[3]| 73.88/78.89  | 70.65/76.43   |  65.23/71.89     |
> >|InfLoRA[4]| 76.95/81.81| 74.75/80.67 | 69.89/76.68 |
> >|S-LoRA|**79.15**/**83.01**  | **77.34**/**82.04** | **75.26**/**80.22**|
> >
> > [1] McDonnell, Mark D., et al. "Ranpac: Random projections and pre-trained models for continual learning." NeurIPS, 2024. \
> > [2] Zhou, Da-Wei, et al. "Revisiting class-incremental learning with pre-trained models: Generalizability and adaptivity are all you need." IJCV, 2024.\
> > [3] Wang, Xiao, et al. "Orthogonal Subspace Learning for Language Model Continual Learning." EMNLP, 2023.\
> > [4] Liang, Yan-Shuo, and Wu-Jun Li. "InfLoRA: Interference-Free Low-Rank Adaptation for Continual Learning." CVPR, 2024.
>
> **Q3**: The two efficient versions of S-LoRA seem unrelated to the core idea of this paper, and the effect is not significant.
> > We respectfully emphasize the critical significance of the two efficient versions of S-LoRA in ensuring that our proposed core idea of S-LoRA is not only effective but, more importantly, **practically efficient**.
> > - As discussed in our response to W1 & Q1, current LoRA-based CL methods typically treat all LoRA components equally, incrementally adding them as new tasks are introduced. However, this training approach becomes increasingly inefficient as the number of tasks grows.
> > - Based on the findings and theoretical analysis of S-LoRA, we have shown that the LoRA components learned later are less important within our framework. This key insight forms the foundation for our proposed ES-LoRA, an efficient variant of S-LoRA. Therefore, these two versions are directly tied to the core idea of S-LoRA,  **focusing on improving efficiency (see updated Table 5) without compromising performance (see Tables 2-3)**.

---

> > ### Comment · Reviewer_yo9h · 2024-11-22
> >
> > Your response has sufficiently addressed my concerns, so I have revised the scores. Your revised paper should further compare or discuss the reference [1-8].
> >
> > **Reference**
> >
> > [1] Liu J, Wu J, Liu J, et al. Learning Attentional Mixture of LoRAs for Language Model Continual Learning[J]. arXiv preprint arXiv:2409.19611, 2024.
> >
> > [2] Dou S, Zhou E, Liu Y, et al. LoRAMoE: Alleviating World Knowledge Forgetting in Large Language Models via MoE-Style Plugin[C]//Proceedings of the 62nd Annual Meeting of the Association for Computational Linguistics (Volume 1: Long Papers). 2024: 1932-1945.
> >
> > [3] McDonnell, M. D.; Gong, D.; Parvaneh, A.; Abbasnejad, E.; and van den Hengel, A. 2024. Ranpac: Random projections and pre-trained models for continual learning. Advances in Neural Information Processing Systems, 36.
> >
> > [4] Zhou, D.-W.; Ye, H.-J.; Zhan, D.-C.; and Liu, Z. 2023b. Revisiting class-incremental learning with pre-trained models: Generalizability and adaptivity are all you need. arXiv preprint arXiv:2303.07338.
> >
> > [6] Xiao Wang, Tianze Chen, Qiming Ge, Han Xia, Rong Bao, Rui Zheng, Qi Zhang, Tao Gui, and Xuanjing Huang. 2023a. Orthogonal subspace learning for language model continual learning. arXiv preprint arXiv:2310.14152.
> >
> > [7] Yang S, Ning K P, Liu Y Y, et al. Is Parameter Collision Hindering Continual Learning in LLMs?[J]. arXiv preprint arXiv:2410.10179, 2024.
> >
> > [8] Continual Learning with Pre-Trained Models: A Survey.

---

### Official Review · Reviewer_6nN3 · 2024-11-03

**Soundness:** 3
**Presentation:** 3
**Contribution:** 3
**Rating:** 8
**Confidence:** 5

**Summary:**

The authors propose to incrementally learn how to merge different LoRA models by keeping old LoRA modules fixed and only update their weights and a new LoRA module in the memory-free CL scenario. The authors compare their method S-LoRA as well as variants of it on multiple benchmarks and different benchmarks against multiple baselines.

**Strengths:**

The authors present a simply, but principled idea. It makes a lot of sense to learn to fuse models in the context of CL.

Besides the choices mentioned in "Weaknesses", the empirically evaluation is convincing.

**Weaknesses:**

The presented work uses ideas from model merging/fusing, e.g., "Editing Models with Task Arithmetic" by Ilharco et al., but doesn't discuss these works in the related work. Adoption of these ideas to CL, such as "Task Arithmetic with LoRA for Continual Learning", are not discussed or not compared to.

The authors compare apples with oranges. Most baselines use inferior PEFT methods (prompt tuning) while the authors use LoRA which naturally will give them an edge. It would be great if the authors could show that the improvement over the baseline is not coming from the PEFT method. See a discussion on this topic in "Choice of PEFT Technique in Continual Learning: Prompt Tuning is Not All You Need".

Table 1 mixes desired properties of CL methods with general properties of methods. End-to-end optimization is no desired property, high predictive performance is.

**Questions:**

N/A

---

> ### Author Response · Authors · 2024-11-21
> **To Reviewer 6nN3 (Part I)**
>
> We sincerely thank the reviewer for providing valuable feedback. We detail our response below point by point. Any modifications made to the paper are highlighted in blue for your convenience. Please kindly let us know whether you have any further concerns.
>
> **W1**: The presented work uses ideas from model merging/fusing, but doesn't discuss them in the related work.
> > Thanks for your insightful suggestion. Following your advice, we have incorporated a discussion on model merging/fusion into the related work section, highlighted in blue in the revised paper. For your convenience, we also detail **the distinct differences between model merging/fusion works and our proposed S-LoRA** below, while we agree with the reviewer that model merging and fusion concepts are insightful in the context of continual learning.
> >   - We first summarize the **key idea of the works [1,7] that adapt model merging for CL**.
> >      - For clarity, let $\theta_0$ represent the foundation model's weights, and $\theta^*_i$ denote the fine-tuned weights of the foundation model on the $i$-th task.
> >      - The original objective of model merging is to fuse the well-learned $\theta_i^*$ or the task vectors $\tau_i=\theta_i^*-\theta_0$ of all tasks, thereby being well-suited for improving the performance of all tasks under multi-task learning.
> >      - When adapted for continual learning, as in [1, 7], each incoming task is fine-tuned sequentially, resulting in its own task vector. At the end of the sequence, all task vectors are merged to mitigate forgetting. Specifically, [1] efficiently fine-tunes a LoRA component for each task, which is regarded as a representation of a task vector (i.e., $\tau_j = \theta^*_j-\theta_0 = {\bf{A}}_j {\bf{B}}_j$). At the end of the sequence, all LoRAs are weighted and merged to $\theta_0$ following Eqn. (5)(6) in [1].
> >   - Our proposed S-LoRA distinguishes it from the above model merging-based works [1] in **three unique contributions**:
> >      - **(C1) Unique contribution 1:** We follow the conventional CL setup, where the model to fine-tune for task $j$ is $\theta^*_{j-1}$ that has been fine-tuned on the previous task $j-1$, instead of $\theta_0$. Thus, the LoRAs in S-LoRA are **incrementally learned on top of the model weights from the previous task**, compared to those LoRAs in model merging-based works that learned on top of the foundation model and thus can be regarded as task vectors.
> >     - **(C2) Unique contribution 2:** The proposed S-LoRA **decouples the magnitude and direction** of the learned LoRA components during sequential training (i.e., $\Delta W = \\|AB\\|\cdot\overline{AB}$; see Lines 185–194 in the paper). This novel design is thoroughly validated in our experiments and has been recognized by both Reviewers on1i and Uver. In contrast, model merging methods adapted to CL typically use the learned LoRA directly to represent the task vector, without such decoupling.
> >      - **(C3) Unique contribution 3:** We theoretically and empirically demonstrate that in our proposed ES-LoRA, **the rank of later-learned LoRA components can be reduced to enhance efficiency**. In contrast, model-merging approaches adapted to CL typically treat all tasks or vectors as equally important, often assigning the same rank to all their learned LoRA components.
> > - To further address the reviewer's concern, we have implemented the paper "*Task Arithmetic with LoRA for Continual Learning* [1]" and provided the empirical comparisons below.
> >   - For a fair comparison with ours in the same number of extra parameters, we set the rank of all LoRAs in [1] to 10. Besides, we configure the memory buffer, which is required by [1] to fine-tune the final merged model, to store 20 samples from each task.
> >   - **Effectiveness of (C1)**: In the table below, we provide the results of two versions of [1], where `v1` exactly follows [1] with a LoRA as a task vector $\\tau_j=\\theta_j^*-\\theta_0$ and $\\theta_j^*$  fine-tuned from $\theta_0$ and `v2` **adapts [1] with our unique contribution 1 C1 equipped**, i.e., a task vector is $\\tau_j=\\theta_j^{\ast'} - \theta_0$   with $\\theta_j^{\ast'}$ obtained via merging the LoRA fine-tuned on task $j$ from $\\theta_{j-1}^{\ast}$ to $\\theta_{j-1}^{\ast}$. The significant performance improvement of `v2` and our S-LoRA not involving subtraction of the foundation model $\theta_0$ over `v1` proves the effectiveness of C1.
> >   - **Effectiveness of (C2)**: In Table 4, S-LoRA shows superiority over its ablated version with no decoupling of previous LoRAs and update of magnitudes of them, confirming the effectiveness of C2.
> >   - **Effectiveness of (C3)**: We show in the updated Table 5 that ES-LoRA equipped with C3 improves efficiency over S-LoRA without compromising performance (see Tables 2-3).

---

> ### Author Response · Authors · 2024-11-21
> **To Reviewer 6nN3 (Part II)**
>
> >| Methods| Equation |IN-R(N=5)(Acc/AAA)|IN-R(N=10)(Acc/AAA)|
> >|:-:|:-:|:-:|:-:|
> >|Task Arithmetic LoRA[1] (`v1`) |$W_0+\sum_{i=1}^N\tau_i$  |   67.11/73.67 | 63.14/70.19    |
> >|Task Arithmetic LoRA[1] (`v2`) |$W_0+\sum_{i=1}^N \tau_i'$  |  72.77/78.07 | 71.02/76.89
> >|S-LoRA|$W_0+\sum_{i=1}^N\alpha_i\overline{A_iB_i}$ |**79.15/83.01**  | **77.34/82.04** |
> >
> **W2**: Most baselines use inferior PEFT methods (prompt tuning) while the authors use LoRA which naturally will give them an edge. It would be great if the authors could show that the improvement over the baseline is not coming from PEFT method.
> > Thank you for your insightful comment. To demonstrate that the improvement of our proposed S-LoRA is not solely attributable to our choice of LoRA over prompts,
> >  -  we have already compared S-LoRA with **the current SOTA LoRA-based method, InfLoRA (CVPR'24) [4]**, in the experimental section (see Table 2 & 3);
> > -  we also included additional LoRA-based methods  in our evaluation during the response period, with the results presented in the following table.
> >    - Compared to other LoRA-based methods [1-4], S-LoRA **achieves the best performance** across various task sequence lengths. This demonstrates that the observed performance gains are **not simply due to replacing prompts with LoRA**. For instance, O-LoRA [2] applies orthogonal regularization to parameters rather than directly addressing the feature space, thus limiting its effectiveness in mitigating forgetting. Task Arithmetic LoRA [1] simply adding task vectors ($\tau_i$) ignores relationships between tasks. As task sequence length increases, the accumulation of conflicting task vectors leads to performance degradation.
> >    - Compared to O-LoRA[2], which treats all LoRA components equally and continually adds them, our proposed ES-LoRA reduces the ranks of later-added LoRAs and thus **improves efficiency without sacrificing performance**. Such rank reduction is supported by our theoretical and empirical findings that the LoRA components learned later contribute less within our framework.
> >
> >| Methods| IN-R(N=5)(Acc/AAA)|IN-R(N=10)(Acc/AAA)|IN-R(N=20)(Acc/AAA)|
> >|:-|:-:|:-:|:-:|
> >|Task Arithmetic LoRA[1]| 72.77/78.07 | 71.02/76.89     | 63.29/70.88    |
> >|O-LoRA[2]| 73.88/78.89  | 70.65/76.43   |  65.23/71.89     |
> >|InfLoRA[4]| 76.95/81.81| 74.75/80.67 | 69.89/76.68 |
> >|S-LoRA|**79.15**/**83.01**  | **77.34**/**82.04** | **75.26**/80.22|
> >|ES-LoRA1| 79.01/82.50 | 77.18/81.74  | 74.05/**80.65**  |
> >
> **W3**: End-to-end optimization is no desired property, high predictive performance is in Table 1.
> > We totally agree with the reviewer that high predictive performance is desired. This is precisely why, in Table 1, we highlight `end-to-end optimization` which has been proved in [6] to strongly correlate with high predictive performance.
> > - CodaPrompt [6] introduces an attention-based component-weight mechanism that allows end-to-end optimization for the first time, distinguishing it from previous works like L2P [5].
> > - As shown in Section 5.3 of [6] (see the table below), this attention mechanism indeed leads to higher average accuracy and less forgetting, underscoring the benefits of end-to-end optimization.
> >
> >In response to your feedback, we have revised Section 1 and marked the changes in blue to enhance clarity and ensure the accuracy of our explanation.
> >
> >| Methods| Acc $\uparrow$|  FM $\downarrow$
> >|:-:|:-:|:-:|
> >|Coda-Prompt[6]| **75.45 $\pm$ 0.56**| **1.64 $\pm$ 0.10**|
> >|Coda-Prompt(w/o End-to-End training)| 74.52 $\pm$ 0.65| 1.67 $\pm$ 0.13|
>
> [1] Chitale, Rajas, et al. "Task Arithmetic with LoRA for Continual Learning." arXiv preprint arXiv:2311.02428 (2023). \
> [2] Wang, Xiao, et al. "Orthogonal Subspace Learning for Language Model Continual Learning." EMNLP, 2023. \
> [3] Liu J, Wu J, Liu J, et al. Learning Attentional Mixture of LoRAs for Language Model Continual Learning. arXiv, Sep.29, 2024.\
> [4] Liang, Yan-Shuo, and Wu-Jun Li. "InfLoRA: Interference-Free Low-Rank Adaptation for Continual Learning." CVPR, 2024.\
> [5] Wang, Zifeng, et al. "Learning to prompt for continual learning." CVPR, 2022.\
> [6] Smith, James Seale, et al. "Coda-prompt: Continual decomposed attention-based prompting for rehearsal-free continual learning." CVPR, 2023.\
> [7] Ilharco, Gabriel, et al. "Editing models with task arithmetic." ICLR, 2023.

---

> > ### Comment · Reviewer_6nN3 · 2024-11-23
> >
> > I'd like to thank the authors for all the effort to address my comments, questions and concerns. I've seen only few edits being made in the paper. It would be great if at least the comparison to task arithmetic LoRA could make it into the final version (Table 4 could be a good place for it). I think these additional studies are very useful and could be interesting for future readers.
> >
> > I have no further questions.

---

### Meta-Review · Area_Chair_ojqn · 2024-12-19

**Metareview:**

This paper investigates class incremental learning for foundation models using LoRA. The authors introduce a novel approach, Scalable LoRA (S-LoRA), which enables the incremental learning of task-specific LoRA and importance parameters while maintaining the optimization directions of previous tasks. The technical innovation is commendable. The experimental evaluations provided in the article are comprehensive. These aspects have been unanimously acknowledged by the reviewers. The paper's presentation is well-organized and straightforward to follow. However, certain aspects of the algorithm's description and the analysis of experimental results were not clearly articulated in the initial version. During the rebuttal phase, the authors effectively addressed the raised concerns. Consequently, I recommend accepting this paper.

**Additional Comments On Reviewer Discussion:**

During the discussion phase, most of the concerns have been effectively addressed. All reviewers expressed positive opinions.

---

### Decision · Program_Chairs · 2025-01-22

Accept (Oral)